# Periodic Skill Discovery

Jonghae Park[1]  Daesol Cho[2]  Jusuk Lee[1]  Dongseok Shim[1]  Inkyu Jang[1]  H. Jin Kim[1]
[1]Seoul National University   [2]Georgia Institute of Technology
bdfire1234@snu.ac.kr

## Abstract

Unsupervised skill discovery in reinforcement learning (RL) aims to learn diverse behaviors without relying on external rewards. However, current methods often overlook the periodic nature of learned skills, focusing instead on increasing the mutual dependence between states and skills or maximizing the distance traveled in latent space. Considering that many robotic tasks—particularly those involving locomotion—require periodic behaviors across varying timescales, the ability to discover diverse periodic skills is essential. Motivated by this, we propose Periodic Skill Discovery (PSD), a framework that discovers periodic behaviors in an unsupervised manner. The key idea of PSD is to train an encoder that maps states to a circular latent space, thereby naturally encoding periodicity in the latent representation. By capturing temporal distance, PSD can effectively learn skills with diverse periods in complex robotic tasks, even with pixel-based observations. We further show that these learned skills achieve high performance on downstream tasks such as hurdling. Moreover, integrating PSD with an existing skill discovery method offers more diverse behaviors, thus broadening the agent's repertoire. Our code and demos are available at https://jonghaepark.github.io/psd

## 1  Introduction

A fundamental observation in nature is that nearly all forms of locomotion are inherently periodic. Rhythmic gaits of quadrupeds, the oscillatory motions of fish, and even human walking patterns share a distinct periodic structure, which can be flexibly modulated across multiple timescales [39, 28, 31]. This inherent periodicity not only enables energy-efficient movement [39, 82] but also provides adaptability under varying conditions [28, 31, 82]. Motivated by this understanding, robotics research has leveraged periodic priors to effectively control complex behaviors in various challenging environments [77, 87, 78, 52, 86, 83, 47].

In contrast, unsupervised skill discovery methods [25, 84, 34, 13, 89, 20, 59, 50, 96]—despite their success in learning diverse behaviors without external reward—have rarely addressed the role of periodicity. They primarily focus on maximizing the mutual information (MI) between skills and states [25, 84, 13, 89, 20, 50] or maximizing state deviation based on a given metric [65–67, 75], both of which encourage state diversity, thereby biasing the learned skills toward discovering *where* to go. However, none of these methods address *how* to behave, which requires modeling the periodic structure of behaviors—especially across multiple timescales.

To address this gap, we propose a novel unsupervised skill discovery objective for learning periodic behaviors, which we call **Periodic Skill Discovery (PSD)**. The main idea of PSD is to train an encoder that maps the state space to a circular latent space, where moving along this circular structure naturally implies repetition—a fundamental property of periodicity. This geometric connection between circular embeddings and periodic behaviors makes our approach both intuitive and effective for capturing periodicity. Specifically, the latent space of PSD is designed to encode temporal distance, so that moving along a larger circle corresponds to a longer period, directly linking latent geometry to actual period (Figure 1).

39th Conference on Neural Information Processing Systems (NeurIPS 2025).

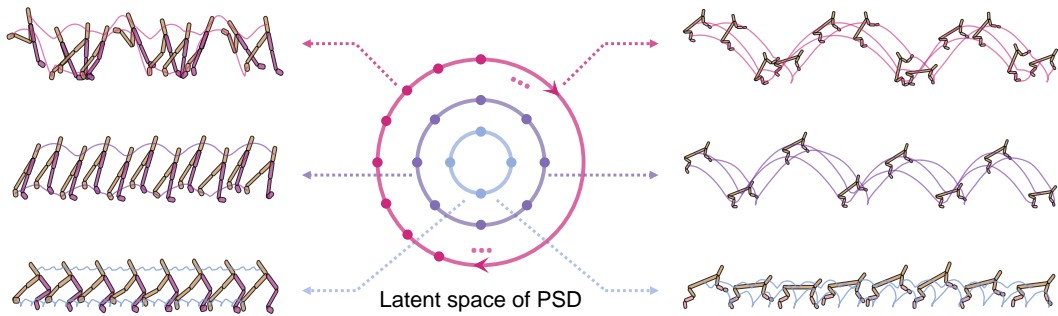

Latent space of PSD

Figure 1: **Visualization of the circular latent space for Walker2D and HalfCheetah.** The core idea of PSD is to map the state space into a circular latent space, where temporal distance is encoded geometrically. The figure visualizes an *actual* policy learned by PSD, where following larger circular paths (*blue* → *magenta*) corresponds to longer-period behaviors.

While the circular representation is being updated, PSD jointly trains an RL policy using a single-step intrinsic reward defined in this latent space. By encouraging the policy to move along the circular path in latent space, the RL agent can achieve periodic skills of varying lengths using only single-step reward signals.

Through experiments on various robotic continuous-control tasks, we empirically demonstrate that PSD can discover diverse periodic skills across multiple timescales. These learned skills are also shown to be effective in solving complex downstream tasks that require multi-timescale prediction (e.g., hurdling). Furthermore, since PSD encodes temporal distance in a manner that is invariant to the underlying state representation [94, 67, 68], it can also discover periodic skills even in pixel-based robotic environments. Moreover, PSD can be effectively combined with the existing skill discovery method, METRA [67], thereby broadening the scope of learned behaviors. We empirically find that this combination leads to more diverse and structured skill repertoires than either method alone.

To sum up, our contributions can be summarized as follows:

- We introduce PSD, a novel skill discovery objective that learns periodic behaviors across multiple timescales by mapping states to a circular latent space, enabling the agent to exhibit temporally structured behaviors with controllable periodicity.

- The discovered skills are predictive over multiple horizons, enabling agents to solve complex downstream tasks (e.g., hurdling) more effectively.

- By encoding temporal distance rather than relying on specific state representations, PSD can discover various periodic behaviors even in pixel-based environments.

- PSD can be combined with the existing skill discovery method, METRA [67], expanding the range of learnable behaviors.

## 2    Related Work

**Learning Periodic Motion**    Recent research has proposed various approaches to learning periodic motion in robotics. In the domain of legged robots, conventional methods often rely on carefully designed foot contact schedules [4, 8, 7] or central pattern generators (CPGs) [77, 87, 78] to manage gait patterns. In RL-based approaches, hand-crafted reward functions [52, 86, 83, 2] or constraints [47] are widely used to encourage specific gait behaviors. These reward functions often incorporate phase variables [8, 52, 86] to inform the current gait phase, or leverage predefined foot trajectories [83, 98, 58, 2] to establish joint targets via inverse kinematics. While effective in guiding legged robots to achieve desired walking patterns, these approaches present significant limitations in terms of generalizability and scalability. Designing such reward functions requires extensive manual tuning and domain-specific knowledge, making it challenging to expand these methods to a wide range of robotic platforms or high-dimensional observations.

Another line of research in learning structured periodic motion focuses on representing motion data using frequency-domain features [57, 11, 92, 5, 97, 88]. In particular, PAE [88] leverages Fourier transforms to encode motion data into a latent phase space, capturing nonlinear local periodicities across different body segments and enabling structured motion representations. Building upon this, FLD [56] introduced an RL stage to PAE, proposing a robust policy learning framework that generates periodic behaviors over long-term horizons. Despite its contributions, FLD relies on offline data to pre-train the autoencoder, limiting its applicability to the given data distribution. Furthermore, it requires manually engineered reward functions for individual body segments, which hinders its scalability to high-dimensional inputs such as pixel-based observations.

**Mutual Information-based Skill Discovery**    A widely adopted approach to unsupervised skill discovery is to learn skills that maximize the mutual information (MI) between states $S$ and skill $Z$, namely $I(S; Z)$. By maximizing $I(S; Z)$, each distinct skill variable $z$ corresponds to distinguishable states $s$, which encourages skill policy to visit a diverse set of states. For example, DIAYN [25] maximizes a variational lower bound of $I(S; Z)$ through the following objective:

$$I(S; Z) = -H(Z \,|\, S) + H(Z) = \mathbb{E}_{z,\tau}[\log p(z \,|\, s)] - \mathbb{E}_z[\log p(z)] \tag{1}$$

$$\geq \mathbb{E}_{z,\tau}[\log q_\theta(z \,|\, s)] + (\text{constant}) \simeq \mathbb{E}_{z,\tau}\left[-\frac{1}{2\sigma^2}\|z - \mu_\theta(s)\|_2^2\right] + (\text{constant}), \tag{2}$$

where $q_\theta(z \,|\, s)$ is a *skill discriminator* that infers the skill $z$ from a given state $s$. The agent is rewarded whenever it visits a state where the discriminator can predict the skill with high confidence.

However, MI-based methods tend to discover skills that are easy to distinguish, rather than skills with diverse temporal patterns. The objective can be fully optimized simply by making the visited states *maximally separated* for each skill (i.e., minimizing $H(Z \,|\, S)$), often leading to simple or static behaviors as there is no additional motivation for exploration [89, 50, 66, 67]. Moreover, when $q_\theta(z \,|\, s)$ is parameterized as a Gaussian $\mathcal{N}\big(\mu(s), \sigma^2 I\big)$, the MI objective can be viewed as a goal-reaching objective in the latent space as shown in Eq. (2) [20, 65]. Consequently, MI-based skill discovery methods do *not* consider periodic nature of behaviors, leaving temporal aspects of skills underexplored.

**Distance-Maximizing Skill Discovery**    As an alternative to MI-based approaches, distance-maximizing methods have been proposed [65–67, 75]. Formally, they maximize the following objective:

$$\mathcal{J}_{\text{DSD}} := \mathbb{E}_{(z,\tau)\sim\mathcal{D}}\left[(\phi(s_{t+1}) - \phi(s_t))^\top z\right] \quad \text{s.t.} \quad \|\phi(x) - \phi(y)\| \leq d(x, y) \quad \forall x, y \in \mathcal{D}, \tag{3}$$

where $\mathcal{D}$ is the replay buffer, and $\phi : \mathcal{S} \to \mathcal{Z}$ is a trainable function that maps states into latent representations. Here, the metric $d$ enforces an upper bound on latent transitions so that differences in the latent space do not exceed the distance measured by $d$. Under this constraint, the RL agent learns to maximize $\|\phi(s_{t+1}) - \phi(s_t)\|$ in certain directions $z$, thereby discovering diverse skills that traverse the largest distances in latent space. Specifically, different choices of the metric $d$—such as Euclidean [65], controllability-aware distance [66], temporal distance [67], and language-based distance [75]—encourage different types of behavioral diversity.

However, a key limitation of distance-maximizing approaches is that they discover skills which *maximally* deviate under their own metrics, yielding only "hard-to-achieve" behaviors. For instance, METRA [67] employs a temporal distance as its metric and thus strongly prefers fast-moving skills to maximize temporal state deviations. This suggests that these distance-maximizing approaches provide *no* incentive to adjust the temporal patterns of the learned skills, making it difficult to capture multi-timescale periodic behaviors.

**Advantages of PSD**    Prior approaches in robotics often require extensive domain-specific knowledge or offline data to learn periodic motion, while unsupervised skill discovery methods fail to capture the periodic structure of behaviors. To overcome these limitations, our proposed method, PSD, constructs a circular latent space that captures multi-timescale periodicity in an unsupervised manner. Moreover, by encoding temporal distance in the latent space, PSD becomes invariant to the underlying state representation and scales to high-dimensional observations. Overall, PSD offers a generalizable and scalable framework for capturing multi-timescale periodicity, enabling RL agents to autonomously achieve periods of diverse lengths.

# 3 Periodic Skill Discovery

In this section, we describe an objective designed to learn circular latent representations that capture periodicity. Leveraging this latent structure, we define intrinsic reward functions to train a skill policy that discovers periodic behaviors.

## 3.1 Preliminaries

For unsupervised skill discovery, we consider a Markov decision process (MDP) $\mathcal{M} \equiv (\mathcal{S}, \mathcal{A}, \mathcal{P})$ in the absence of external reward. Here, $\mathcal{S}$ is the state space, $\mathcal{A}$ is the action space, and $\mathcal{P} : \mathcal{S} \times \mathcal{A} \to \Delta(\mathcal{S})$ denotes the transition function. In this setup, we define a positive integer $L$ as the *period variable*, which conditions the policy $\pi(a \,|\, s, L)$ to produce behaviors with period $2L$. Formally, we refer to $\pi(a \,|\, s, L)$ as a *periodic skill* policy, which satisfies

$$s_t = s_{t+2L} \quad \text{where } \{s_{t+k}\}_{k=0}^{2L} \sim \mathcal{P}_{\pi_L} \quad \forall t \in \{0, 1, \dots\}.$$

Here, $\mathcal{P}_{\pi_L}$ denotes the distribution over state trajectories induced by the policy $\pi(a \,|\, s, L)$. At the beginning of each training episode, the period variable $L$ is sampled from a prior distribution $p(L)$, which we assume to be uniform over a bounded set of positive integers $L \in [L_{\min}, L_{\max}]$. Once sampled, $L$ remains fixed throughout the episode. We then roll out the periodic skill policy $\pi(a \,|\, s, L)$ using the chosen $L$ to collect a skill trajectory.

## 3.2 Circular Latent Representation to Capture Periodicity

To capture periodicity in an unsupervised manner, we train an encoder $\phi : \mathcal{S} \times \mathbb{N} \to \mathbb{R}^d$ that maps a state $s$ and a period variable $L$ to a latent circle of diameter $L$. For simplicity, we denote $\phi_L(s) := \phi(s, L)$ so that $\phi_L(\cdot)$ highlights the dependence on $L$. Formally, PSD maximizes the following constrained objective:

$$\mathcal{J}_{\text{PSD}} := \mathbb{E}_{(L, s_t, s_{t+L}) \sim \mathcal{D}} \Big[ \|\phi_L(s_{t+L}) - \phi_L(s_t)\|_2 - k \,\|\phi_L(s_{t+L}) + \phi_L(s_t)\|_2 \Big] \tag{4}$$

$$\text{s.t.} \quad \|\phi_L(s_{t+L}) - \phi_L(s_t)\|_2 \;\leq\; L, \tag{5}$$

$$\|\phi_L(s_{t+1}) - \phi_L(s_t)\|_2 \;\leq\; L \sin\big(\pi/2L\big) \quad \forall (L, s_t, s_{t+1}, s_{t+L}) \in \mathcal{D}, \tag{6}$$

where $\mathcal{D}$ is the replay buffer and $k > 0$ is a constant.

To construct a circular latent representation, $\mathcal{J}_{\text{PSD}}$ encourages the encoder $\phi_L$ to map $s_t$ and $s_{t+L}$ to opposite points of the latent circle of diameter $L$. This is achieved by maximizing $\|\phi_L(s_{t+L}) - \phi_L(s_t)\|_2$ while the first constraint ensures that this distance does not exceed $L$. The second constraint enforces equal angular spacing between consecutive states, where each adjacent pair is separated by an angle of $\pi/L$, resulting in a regular arrangement along the circle. Specifically, the term $L \sin(\pi/2L)$ corresponds to the side length of a *regular* $2L$-gon inscribed in a circle of diameter $L$. As a result, the encoded states are positioned at the vertices of the polygon, evenly distributed along the circular latent space (Figure 2), which facilitates the design of a single-step intrinsic reward described in Section 3.3.

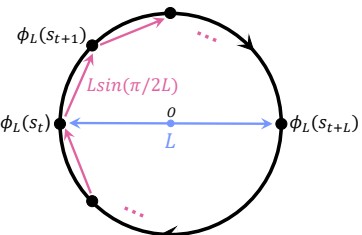

Figure 2: **Latent space of PSD.** Illustration of the circular structure induced by optimizing $\mathcal{J}_{\text{PSD}}$.

Additionally, to prevent arbitrary translations in the latent space, we include the term $-k \,\|\phi_L(s_{t+L}) + \phi_L(s_t)\|_2$ in Eq. (4). This ensures that the midpoint of opposite points is placed at the origin, aligning circles of different diameters to share the same center and form concentric circles for each $L$.

By optimizing $\mathcal{J}_{\text{PSD}}$, the latent representation is structured to capture temporal distances. States that are $L$ steps apart are mapped to opposite points on the latent circle, and after $2L$ steps, the latent trajectory returns to its initial point, completing a full loop. We formally prove in Appendix A that optimizing $\mathcal{J}_{\text{PSD}}$ induces a *regular* $2L$-gon in latent space, where the encoded states satisfy $\phi_L(s_t) = \phi_L(s_{t+2L})$ and $\phi_L(s_{t+L})$ lies opposite to $\phi_L(s_t)$ on a circle of diameter $L$.

---

**Algorithm 1** Periodic Skill Discovery (PSD)

---

1: **Initialize**: policy $\pi$, encoder $\phi$, sampling bound $L_{\min,\max}$, replay buffer $\mathcal{D}$, Lagrange multiplier $\lambda$
2: **for** each training epoch **do**
3:    Update $L_{\min}$, $L_{\max}$ **if** *AdaptiveSampling* is enabled
4:    **for** each episode in the epoch **do**
5:       Sample $L \sim p(L)$ where $L \in [L_{\min}, L_{\max}]$
6:       Execute $\pi(a \,|\, s, L)$ for the entire episode, and store transitions $(L, s_t, a_t, s_{t+1})$ in $\mathcal{D}$
7:    **end for**
8:    Update $\phi_L(s)$ by maximizing $\mathcal{J}_{\mathrm{PSD},\phi}$ using samples from $\mathcal{D}$
9:    Compute intrinsic reward $r_{\mathrm{PSD}}$
10:    Update $\pi(a \,|\, s, L)$ with $r_{\mathrm{PSD}}$ using SAC
11: **end for**

---

**Tractable Implementation**   To implement our constrained objective $\mathcal{J}_{\mathrm{PSD}}$ in a tractable manner, we use dual gradient descent [9, 24] with Lagrange multipliers $\lambda_{1,2} \geq 0$ as follows:

$$
\begin{aligned}
\mathcal{J}_{\mathrm{PSD},\phi} = \mathbb{E}\Big[ &\|\phi_L(s_{t+L}) - \phi_L(s_t)\|_2 - k \,\|\phi_L(s_{t+L}) + \phi_L(s_t)\|_2 \\
&+ \lambda_1 \cdot \min\big(\epsilon,\, L - \|\phi_L(s_{t+L}) - \phi_L(s_t)\|_2\big) \\
&+ \lambda_2 \cdot \min\big(\epsilon,\, L\sin(\pi/2L) - \|\phi_L(s_{t+1}) - \phi_L(s_t)\|_2\big)\Big],
\end{aligned}
\tag{7}
$$

where $\epsilon > 0$ is a small relaxation constant introduced to improve training stability [94, 67]. The tuple $(L, s_t, s_{t+1}, s_{t+L})$ is sampled from the replay buffer, which stores trajectories collected by the skill policy $\pi(a \,|\, s, L)$.

### 3.3   Single-Step Transition Reward for Periodic Behavior

While a circular representation is being learned, the RL agent is jointly trained with an intrinsic reward that encourages periodic behavior. Since the circular latent space is designed to capture periodicity, rewarding the policy for moving along this circular space naturally promotes the learning of periodic behaviors. To this end, we first quantify how much a single-step latent transition deviates from the optimal length:

$$
\Delta \;:=\; \|\phi_L(s_{t+1}) - \phi_L(s_t)\|_2 \;-\; L\sin\big(\pi/2L\big).
$$

Here, $L\sin\big(\pi/2L\big)$ is the optimal single-step length from Eq. (6). We then define the intrinsic reward $r_{\mathrm{PSD}}$ as follows:

$$
r_{\mathrm{PSD}}(s_t, s_{t+1}, L) := \exp\big(-\kappa\,\Delta^2\big),
\tag{8}
$$

where $\kappa > 0$ is a positive constant. Maximizing $r_{\mathrm{PSD}}$ penalizes deviation from the optimal single-step distance in the latent space, thereby encouraging the policy $\pi(a \,|\, s, L)$ to follow the circular path and complete a full cycle of period $2L$, where $\phi_L(s_t) = \phi_L(s_{t+2L})$. By leveraging the latent representation of PSD, the RL agent can discover diverse skills with varying periods using only a single-step reward design, without requiring entire rollouts or specialized objectives for each period.

### 3.4   Adaptive Sampling Method

To enable the agent to discover a *maximally* diverse range of periods without any prior knowledge of its inherent period ranges, we introduce an adaptive sampling method that dynamically adjusts the sampling range $[L_{\min}, L_{\max}]$ during training. The idea is to evaluate the performance of the policy conditioned on the boundary of the current sampling range. The performance is measured by the average cumulative sum of $r_{\mathrm{PSD}}$ as follows:

$$
\bar{R}_L = \mathbb{E}_{p(\tau|L)}\Big[ \sum_{t=0}^{T-1} r_{\mathrm{PSD}}(s_t, s_{t+1}, L) \Big] \;\; \text{for } L \in \{L_{\min}, L_{\max}\},
$$

where $p(\tau \,|\, L)$ denotes the distribution over state trajectories induced by the policy $\pi(a \,|\, s, L)$. Notably, since the $r_{\mathrm{PSD}}$ is defined as $\exp(-\kappa\Delta^2) \in (0, 1]$, $\bar{R}_L$ is upper bounded by $T$, which corresponds to the maximum episode length. We use this upper bound to set a threshold for how accurately the

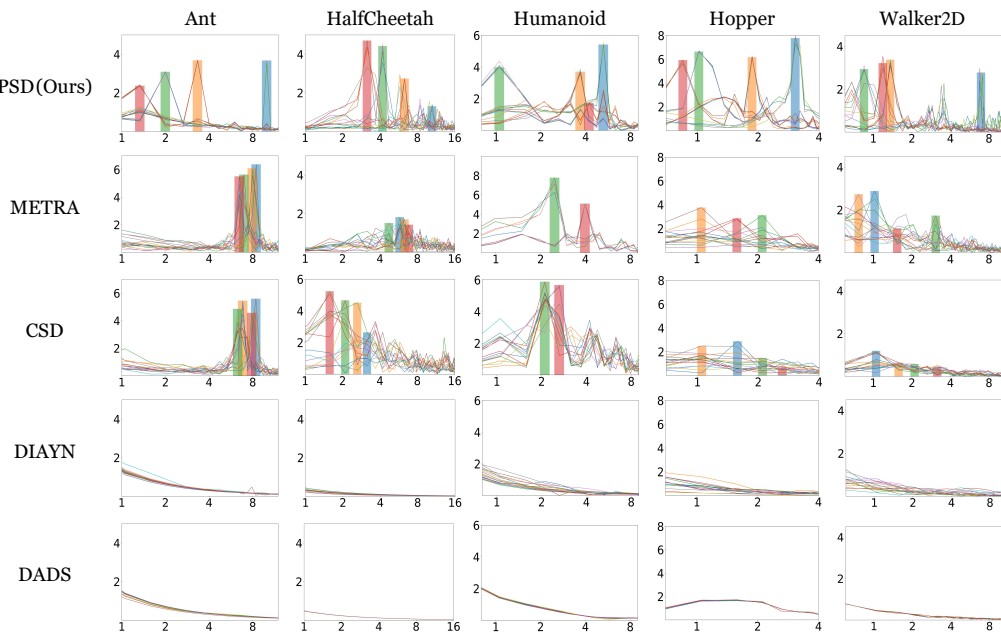

Figure 3: **Comparison of skill trajectories in the frequency domain.** We apply a Fourier transform to skill trajectories, where each skill is uniformly sampled from the skill prior of each method. The resulting spectrum illustrates the frequency ($x$-axis) and amplitude ($y$-axis), representing the temporal patterns of each skill. The accompanying bar chart visualizes the four most dominant frequencies—ranked by amplitude—and highlights the range of discovered periods.

policy $\pi(a \mid s, L)$ follows the desired circular path in the latent space. The bounds are updated as follows:

$$L_{\max} \leftarrow \begin{cases} L_{\max} + N & \text{if } \bar{R}_{L_{\max}} > \alpha T \\ L_{\max} - N & \text{if } \bar{R}_{L_{\max}} < \beta T \end{cases} \qquad L_{\min} \leftarrow \begin{cases} L_{\min} - N & \text{if } \bar{R}_{L_{\min}} > \alpha T \\ L_{\min} + N & \text{if } \bar{R}_{L_{\min}} < \beta T \end{cases}$$

Here, $\alpha$ and $\beta$ are threshold coefficients, where $\alpha > \beta > 0$, and $N$ is a positive integer that determines the step size for adjusting the bounds. Since $r_{\text{PSD}}$ quantifies the deviation between the optimal and actual latent transitions, a large $\bar{R}_L$ indicates that the current skill policy has the capability to achieve the currently given period ranges and is ready to expand its skills. In such cases, the corresponding bound is extended. Conversely, if $\bar{R}_L$ is too small, the current bound is rejected and the previous value is restored. This mechanism enables the agent to discover dynamically feasible period bounds, thereby broadening the range of achievable periods. Details of the full algorithm and hyperparameters are provided in Appendix C.

### 3.5 Algorithm Summary

To summarize, we train the encoder $\phi$ to construct the circular latent representation, and jointly optimize the policy $\pi(a \mid s, L)$ with the single-step intrinsic reward $r_{\text{PSD}}$ using SAC [33]. The full procedure is described in Algorithm 1, and additional implementation details are provided in Appendix C.

## 4 Experiments

The main goal of our experiments is to demonstrate that PSD can discover diverse periodic skills across multiple timescales by learning a circular latent representation. We also evaluate whether the discovered skills are useful for solving downstream tasks. In addition, we examine the scalability of PSD to high-dimensional observations such as pixel inputs. Finally, we explore the potential of combining PSD with existing unsupervised skill discovery methods to enhance the agent's behavioral diversity.

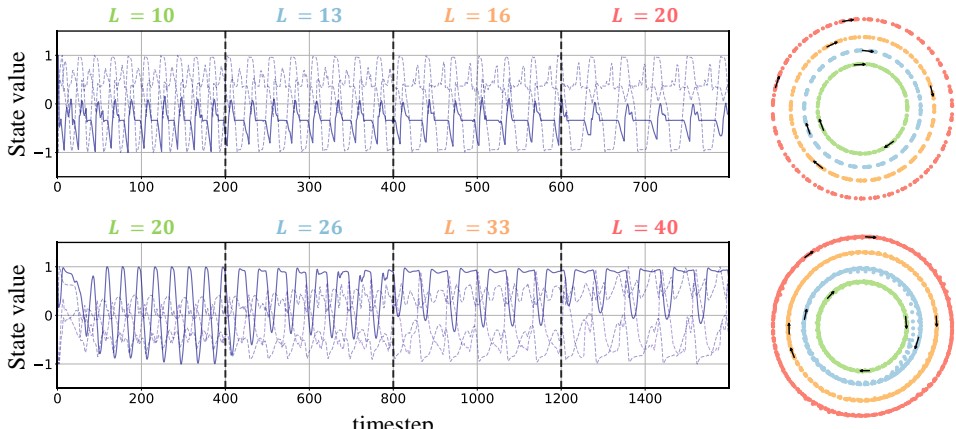

Figure 5: **Trajectories of the skill policy and corresponding latent representation.** The figure shows the joint trajectories of Ant (*top*) and Walker2D (*bottom*) and a 2D PCA projection of their latent encodings learned by PSD. Within a single episode, we *switch* the period variable $L$ at fixed time intervals. The resulting behavior of the skill policy exhibits a period of $2L$ timesteps.

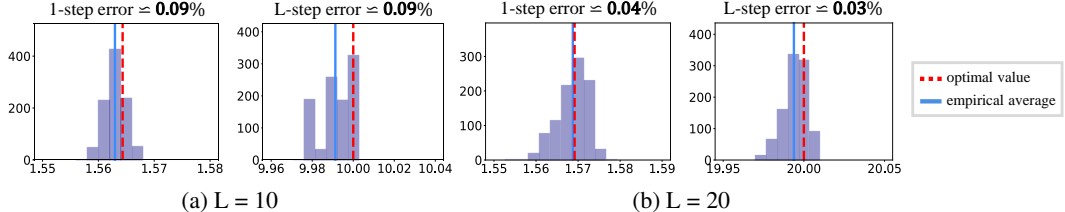

Figure 6: **Histogram of the representation learned by PSD in the Ant environment.** The average values of 1-step distance $\|\phi_L(s_t) - \phi_L(s_{t+1})\|$ and $L$-step distance $\|\phi_L(s_t) - \phi_L(s_{t+L})\|$ converge to their optimal values, indicating that the constraints of the objective $\mathcal{J}_{\text{PSD}}$ are effectively satisfied.

**Experimental Setup**    We evaluate PSD on five robotic locomotion tasks in the MuJoCo environment [10, 91], both in state and pixel domain: Ant, HalfCheetah, Humanoid, Hopper, and Walker2D (Figures 4 and 7).

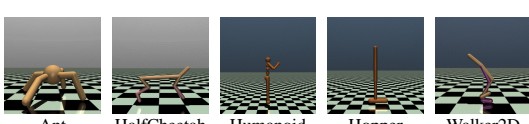

Figure 4: **MuJoCo locomotion environments.**

**Baselines**    We compare PSD with the four state-of-the-art unsupervised skill discovery methods: **DIAYN** [25] is a mutual information-based method that discovers skills by training a skill discriminator $q_\theta(z \mid s)$ to infer the skill from a given state. **DADS** [84], similar to DIAYN, trains skill dynamics $q_\theta(s' \mid s, z)$ to increase the mutual dependence between state transitions and the skill, enabling the agent to learn diverse state transitions conditioned on the skill variable $z$. **CSD** [66] and **METRA** [67] fall into the category of distance-maximizing skill discovery methods. These methods discover skills by maximizing the latent distance traveled in a specific direction of the skill vector $z$. Specifically, CSD uses a controllability-aware distance metric, and METRA uses a temporal distance metric.

**Question 1.**    *Can PSD discover diverse periodic skills across multiple timescales?*

We first check whether PSD can learn a circular latent space constructed by the encoder $\phi$, and whether the skill policy $\pi(a \mid s, L)$ actually produces behaviors with a period of $2L$ across different values of $L$. Figure 5 shows the trajectories of representative states along with a 2D PCA projection of the corresponding latent trajectories for the Ant and Walker2D environments. For varying values of $L$, PSD successfully constructs a circular latent space whose diameter is proportional to the period variable $L$, and learns behaviors with the desired period of $2L$. For example, in the Ant environment with $L = 20$, we can observe that the resulting behavior completes approximately five full cycles of period $2L (= 40)$ over 200 timesteps. These results suggest that, by leveraging a circular-shaped latent space, PSD can learn a policy that produces behaviors with controllable periodicity.

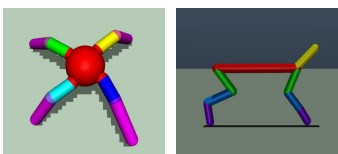
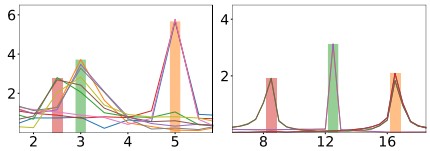

| (a) Pixel-based observation | (b) Frequency spectrum of skill trajectory |
|---|---|

Figure 7: **Frequency spectrum of skills in pixel-based environments.** We visualize the pixel-based observations used as input to PSD, along with the resulting frequency spectrum of skill trajectories obtained via Fourier transform. The accompanying bar chart highlights the top-3 frequency components ranked by amplitude.

Table 1: **Comparison of downstream task performance.** We evaluate PSD against existing skill discovery methods. High-level policies are trained using PPO with the skill policies kept frozen. All reported values are average returns over 10 seeds.

| Downstream task | DIAYN | DADS | CSD | METRA | PSD (Ours) |
|---|---|---|---|---|---|
| HalfCheetah-hurdle | $0.6_{\pm 0.5}$ | $0.9_{\pm 0.3}$ | $0.8_{\pm 0.6}$ | $1.9_{\pm 0.8}$ | $\mathbf{3.8}_{\pm 2.0}$ |
| Walker2D-hurdle | $2.6_{\pm 0.5}$ | $1.9_{\pm 0.3}$ | $4.1_{\pm 1.3}$ | $3.1_{\pm 0.5}$ | $\mathbf{5.4}_{\pm 1.4}$ |
| HalfCheetah-friction | $13.2_{\pm 3.4}$ | $12.4_{\pm 2.9}$ | $12.5_{\pm 3.8}$ | $30.1_{\pm 13.1}$ | $\mathbf{43.4}_{\pm 19.1}$ |
| Walker2D-friction | $4.6_{\pm 1.2}$ | $1.6_{\pm 0.1}$ | $5.3_{\pm 0.3}$ | $5.2_{\pm 1.6}$ | $\mathbf{8.7}_{\pm 1.7}$ |

To quantitatively evaluate whether the learned circular representation satisfies the objective $\mathcal{J}_{\mathrm{PSD}}$, we sample 1k transitions from the replay buffer $\mathcal{D}$ and assess whether the 1-step constraint in Eq. (6) and the $L$-step constraint in Eq. (5) are approximately satisfied. Figure 6 plots histograms of the 1-step distance $\|\phi_L(s_t) - \phi_L(s_{t+1})\|$ and the $L$-step distance $\|\phi_L(s_t) - \phi_L(s_{t+L})\|$ in the circular latent space, computed over sampled transitions. As shown in the figure, both distances converge closely to their theoretical optima, $L\sin(\pi/2L)$ and $L$, with a small relative error. This strong alignment between empirical measurements and analytical predictions indicates that the encoder $\phi$ successfully enforces the geometric regularity of the circular latent space during training. The full experimental results of Figures 5 and 6 are provided in Appendix D.

Next, we compare PSD with prior skill discovery methods—DIAYN, DADS, CSD, and METRA—that aim to learn diverse behaviors via policies of the form $\pi(a \mid s, z)$, conditioned on different skill variables $z$. For each method, we uniformly sample 16 skills from its skill prior and roll them out in the environment to collect corresponding skill trajectories. For comparison, each trajectory is normalized per dimension using statistics computed from random-action rollouts. The normalized trajectories are then projected to a one-dimensional subspace using Principal Component Analysis (PCA). Finally, we apply a Fourier transform to each projected trajectory to analyze its frequency components and extract the four highest frequencies by amplitude, which are visualized as bar charts.

As shown in Figure 3, PSD consistently discovers skills that exhibit a wide range of frequencies due to its explicit modeling of circular periodicity. In contrast, distance-maximizing approaches like METRA and CSD tend to concentrate on narrow frequency bands and often produce inconsistent, indistinguishable behaviors in Hopper and Walker2D, limiting the diversity of the discovered temporal patterns and frequencies. Also, MI-based methods often produce either static or partially random behaviors, as they do not incorporate the temporal aspects of skills.

**Question 2.** *Are the discovered skills useful for solving downstream tasks?*

To evaluate the utility of the discovered skills, we conduct downstream experiments by training a high-level policy $\pi^h(L \mid s)$ (or $\pi^h(z \mid s)$ for baseline methods). For each method, the skill policy is kept frozen, and the high-level policy is trained using PPO [80] to select skills that maximize task-specific rewards. We design downstream environments featuring two types of challenges—hurdles and varying ground friction. In the hurdle task, the agent should select skills to jump over the irregularly placed hurdles, which requires adaptive coordination between multi-timescale skills. Similarly, in the friction task, the agent should select skills to robustly walk across terrain whose surface friction coefficients are randomly assigned. (see Appendix C for details).

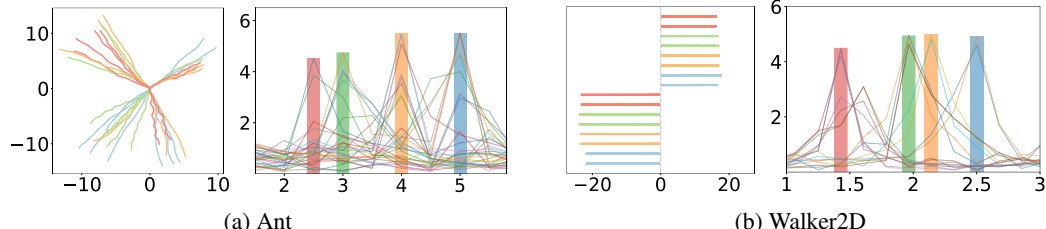

(a) Ant            (b) Walker2D

Figure 8: **Visualization of traveled distance and frequency spectrum of skills learned via the combination of METRA and PSD.** Colors indicate skills conditioned on the same value of the period variable $L$. Videos are available at our project page.

Since our method is not explicitly optimized for exploration in the state space, we add an external velocity-based reward $r_{\text{ext}}$ to encourage forward progress. For fair comparison, the same external reward is linearly combined with the intrinsic rewards of all baseline methods. Table 1 shows that PSD outperforms the baselines on most tasks, demonstrating that PSD provides skills that are both adaptable and robust.

**Question 3.** *Is PSD scalable to high-dimensional observations such as pixel-based input?*

Since PSD encodes periodicity by capturing temporal distances between states, its latent space is invariant to the specific state representation. To validate this, we conducted experiments in pixel-based Ant and HalfCheetah environments, as depicted in Figure 7, and found that PSD successfully learns periodic behaviors even from raw pixel inputs. These results demonstrate that PSD generalizes robustly to visual domains without any modification to its objective or reward formulation, highlighting its scalability to high-dimensional inputs. Additional analyses are provided in Appendix D.

**Question 4.** *Can PSD become fully unsupervised by combining it with other unsupervised methods?*

Since $r_{\text{PSD}}$ is designed to provide additional variations in the learned behaviors, it could be combined with any type of reward, even with an unsupervised one. To validate it, we combine PSD with METRA [67], enabling a fully unsupervised extension. As discussed in Section 2, METRA optimizes the following objective:

$$\max_{\phi_m,\pi} \ \mathbb{E}_{p(\tau,z)}\left[\sum_{t=0}^{T-1}(\phi_m(s_{t+1})-\phi_m(s_t))^\top z\right] \quad \text{s.t.} \ \ \|\phi_m(s')-\phi_m(s)\| \leq 1 \quad \forall(s,s') \in \mathcal{S}_{adj}, \ \ (9)$$

where $\phi_m$ denotes the encoder used in METRA. METRA discovers exploratory skills that maximally deviate along directions $z$ in latent space, while constraining the latent distance between adjacent states to 1, thus capturing temporal distance. PSD naturally aligns with this formulation, as both methods capture *temporal* aspects of skills: METRA adjusts the temporal *direction* of skills (i.e., the skill variable $z$), whereas PSD modulates their temporal *length* (i.e., the period variable $L$). By jointly training both encoders and using the sum of their rewards, we can obtain a skill policy that enables independent control over both variables ($z$ and $L$), as follows:

$$\pi(a\,|\,s,z,L) \leftarrow \arg\max_{\pi} \ \mathbb{E}_{p(\tau,z,L)}\Big[\sum_{t=0}^{T-1}\underbrace{(\phi_m(s_{t+1})-\phi_m(s_t))^\top z}_{r_{\text{METRA}}}+\underbrace{\exp(-\kappa\,\Delta(L)^2)}_{r_{\text{PSD}}}\Big]. \quad (10)$$

In Figure 8, we visualize the traveled XY-coordinates (or X-coordinates) alongside the frequency spectrum of the corresponding skill trajectories. By adjusting the variables $z$ and $L$ of the policy $\pi(a \mid s,z,L)$, the agent can modulate both the movement direction and the period of skills in a fully unsupervised manner, yielding a more diverse behavioral repertoire. This result suggests that the temporal property of PSD is orthogonal to the exploratory objectives of METRA, making it a complementary component for constructing fully unsupervised policies. (see Appendix C for implementation details)

# 5 Conclusion

We introduce **Periodic Skill Discovery (PSD)**, a framework for unsupervised skill discovery that captures the periodic nature of behaviors by embedding states into a circular latent space. By optimizing a constrained objective that encodes temporal distance, PSD enables agents to learn skills with controllable periods. Our experiments demonstrate that PSD discovers diverse and temporally structured skills across various MuJoCo environments, and scales to raw pixel observations. Furthermore, combining PSD with METRA leads to richer behaviors by jointly modulating temporal direction and period. Overall, PSD provides a scalable and principled framework for discovering temporally structured behaviors in reinforcement learning.

**Limitations and Future Work.** While our experiments primarily focus on locomotion tasks, due to their suitability for showcasing multi-timescale behaviors, the PSD framework is applicable to *any* domain that exhibits periodic structure. However, PSD may underperform in settings with large persistent external disturbances (e.g., constant interference from another agent) where periodic behavior becomes infeasible. An interesting future direction is to extend PSD to non-periodic tasks, such as robotic manipulation, by generalizing the latent geometry beyond circular structures. Moreover, directly integrating frequency-domain analysis, such as Fourier representations, into the training process could further improve PSD in capturing temporal patterns.

# Acknowledgments

We would like to thank Kanggyu Park for his invaluable support, and the anonymous reviewers for their insightful comments. This work was supported by Samsung Research Funding & Incubation Center of Samsung Electronics under Project Number SRFC-IT2402-17.

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

## A  Theoretical Results

**Theorem 1.** *Given a positive integer $L$, $\phi_L$ is an optimal solution to $\mathcal{J}_{PSD}$ if and only if it forms a regular $2L$-gon of diameter $L$ centered at the origin.*

*Proof.* We prove the claim by showing both directions of the equivalence.

($\Leftarrow$) Suppose $\phi_L$ forms a regular $2L$-gon of diameter $L$ centered at the origin. Then, for all $(L, s_t, s_{t+1}, s_{t+L}) \in \mathcal{D}$, we have $\|\phi_L(s_{t+L}) - \phi_L(s_t)\|_2 = L$ and $\phi_L(s_{t+L}) = -\phi_L(s_t)$, which implies $\|\phi_L(s_{t+L}) + \phi_L(s_t)\|_2 = 0$. Thus, the objective becomes $\mathcal{J}_{PSD} = \mathbb{E}[L - k \cdot 0] = L$. Since Eq. (5) requires $\|\phi_L(s_{t+L}) - \phi_L(s_t)\|_2 \leq L$, any feasible solution must satisfy $\mathcal{J}_{PSD} \leq L$, and no higher value can be attained. Under this condition, the given regular $2L$-gon satisfies both constraints in Eq. (5) and Eq. (6). Hence, $\phi_L$ is optimal.

($\Rightarrow$) Suppose $\phi_L$ is optimal and achieves $\mathcal{J}_{PSD} = L$. Then $\|\phi_L(s_{t+L}) - \phi_L(s_t)\|_2 = L$ and $\|\phi_L(s_{t+L}) + \phi_L(s_t)\|_2 = 0$, implying $\phi_L(s_{t+L}) = -\phi_L(s_t)$ and $\phi_L(s_t)$ lies on a hypersphere of radius $L/2$ centered at the origin for all $(L, s_t, s_{t+1}, s_{t+L}) \in \mathcal{D}$. From Eq. (6), we have $\|\phi_L(s_{t+1}) - \phi_L(s_t)\|_2 \leq L \sin(\pi/2L)$, which implies that the maximum angular distance between adjacent points is $\pi/L$. Under this condition, reaching the antipodal point $\phi_L(s_{t+L}) = -\phi_L(s_t)$ starting from $\phi_L(s_t)$ is only possible if the points are equally spaced along the circumference of a great circle on the hypersphere, with an angular distance of exactly $\pi/L$ between adjacent points. Hence, $\phi_L$ forms a regular $2L$-gon of diameter $L$ centered at the origin. $\square$

## B  Extended Related Work

PSD primarily falls into the category of unsupervised skill discovery methods [1, 25, 84, 34, 13, 89, 20, 43, 46, 59, 50, 17, 42, 65, 69, 66, 53, 96, 44, 67, 45, 3, 75, 100, 41, 95, 14, 76], which aim to learn a diverse set of skills without external rewards, and the resulting skills can be effectively adapted to downstream tasks or leveraged for high-level planning. In this regard, these methods also share conceptual similarities with Quality-Diversity (QD) algorithms [55, 61, 74, 22, 79, 21, 16, 63, 70, 32, 15], an evolutionary optimization framework that iteratively explores and refines diverse behavioral patterns without external task rewards, using behavior descriptors as implicit objectives to guide exploration. Both frameworks aim to discover a broad repertoire of distinct and high-performing behaviors by optimizing for diversity rather than maximizing a single task-specific reward.

Discovering these diverse and useful repertoires inherently requires an agent to explore the environment broadly and encourage skills to visit a wide range of states. This emphasis on broad exploration shows a strong connection to exploration methods [38, 6, 64, 90, 30, 36, 85, 12, 73, 81, 71, 60, 40, 19, 54, 18] that explicitly aim to maximize state coverage through intrinsic rewards. From this perspective, the PSD framework can also be viewed as exploring the environment in the frequency domain (see Figure 3), as it learns diverse periodic behaviors across multiple timescales in an unsupervised manner through an adaptive sampling method.

Moreover, since PSD learns a latent representation where distances between states capture their temporal relationships, it is closely related to prior works [72, 29, 23, 35, 26, 27, 94, 67, 68, 3, 62, 99] that aim to encode temporal structure in RL representations.

## C  Experimental Details

### C.1  Environments

**MuJoCo locomotion environments**  We adopt MuJoCo environments including Ant, HalfCheetah, Humanoid, Hopper, and Walker2D [10, 91] to evaluate our method and baselines. Episode lengths are set to 200 timesteps for Ant and HalfCheetah, and 400 timesteps for Humanoid, Hopper, and Walker2D. For state-based observations, we follow the default Gym setting [10], which includes proprioceptive information in the observation space. However, since both CSD and METRA rely on global position information to construct their latent representations, we include the global position in the observation when applying these methods, following the setups described in their original papers. For the pixel-based experiments of PSD, we use $90 \times 90 \times 3$ RGB images captured from a tracking camera (view shown in Figure 7) as input to both the RL agent and the encoder $\phi$, without incorporating any additional proprioceptive information.

**Downstream tasks environments**  In the HalfCheetah-hurdle and Walker2D-hurdle environments, the high-level policy receives a reward whenever the agent successfully jumps over a hurdle. For HalfCheetah-hurdle, the hurdle positions are $[2.5, 4.0, 7.0, 10.0, 15.0, 22.0, 30.0]$, with a height of 0.26, which is higher than the setting used in METRA [67]. For Walker2D-hurdle, the hurdle positions are $[1.2, 2.7, 4.1, 5.8, 7.0, 9.2, 11.0, 12.8, 14.2]$, with a height of 0.11. In both environments, the hurdles are unevenly spaced, requiring multi-timescale coordination for successful locomotion. We provide the distance to the nearest hurdle as part of the task-specific input $s_{\text{task}}$ to the high-level policy. The episode lengths are set to 300 timesteps for HalfCheetah-hurdle and 600 timesteps for Walker2D-hurdle.

In the HalfCheetah-friction and Walker2D-friction environments, the agent is rewarded for maintaining forward velocity, which encourages robust locomotion while avoiding falls under changing friction conditions. For implementation simplicity, we do not modify the ground friction directly. Instead, we sequentially change the friction parameters of the agent's feet in the XML file every 100 timesteps, cycling through the values $[0.5, 1.5, 2.0]$, given that the default friction parameter in MuJoCo is 1.0. Additionally, the current friction coefficient is provided as task-specific input $s_{\text{task}}$ to the high-level policy, enabling it to adapt to the changing frictions. The episode lengths are set to 500 for both HalfCheetah-friction and Walker2D-friction.

### C.2  Implementation Details

We implement PSD on top of the publicly available PyTorch SAC implementation[1]. For fair comparison, we implement all baseline methods within the same codebase as PSD to ensure consistency in training procedures and infrastructure. To train the high-level policy for downstream tasks, we use PPO implemented in a public PyTorch repository[2]. All experiments are conducted on an NVIDIA A6000 GPU, and training for each task typically completes within 24 hours.

**Training PSD**  For training PSD, we uniformly sample four discrete values of the period variable $L$ from the range $[L_{\min}, L_{\max}]$, including both bounds, to ensure coverage of the range while maintaining training efficiency. As shown in Appendix D.2, we found that this sampling strategy is sufficient, as the model is able to generalize to intermediate $L$ values via interpolation.

Additionally, since $L$ is a scalar value, directly feeding it into the encoder $\phi$, the policy $\pi$, and the Q-function may limit the representational capacity of these networks. To address this, we apply sinusoidal positional embeddings—commonly used in transformers [93] and diffusion models [37]—to project $L$ into a higher-dimensional space. As an example, rather than using $L$ directly in the policy in the form of $\pi(a \mid s, L)$, we use $\pi(a \mid s, \text{Embed}(L))$, where $\text{Embed}(L)$ denotes the embedded representation of $L$ as follows:

$$\text{Embed}(L) = [e_0, e_1, \ldots, e_{D-1}], \quad \text{where } e_i = \begin{cases} \sin(L \cdot \omega_i) & \text{if } i \bmod 2 = 0, \\ \cos(L \cdot \omega_i) & \text{if } i \bmod 2 = 1, \end{cases}$$

---

[1] https://github.com/pranz24/pytorch-soft-actor-critic
[2] https://github.com/nikhilbarhate99/PPO-PyTorch

where the frequency term $\omega_i$ is defined as $\omega_i = 10000^{-2 \cdot \lfloor i/2 \rfloor / D}$. We apply this sinusoidal embedding to the period variable $L$ whenever it is used as input to the network, enabling the model to better distinguish and generalize across different temporal scales. In addition, we found that using fixed values for $\lambda_1$ and $\lambda_2$ works well in practice, when optimizing $\mathcal{J}_{\text{PSD}}$ via dual gradient descent method. The full set of hyperparameters is summarized in Table 2.

**Training baseline methods**   For baseline methods, we closely followed the implementation details described in their original papers. For METRA, we use a 2-dimensional continuous skill vector $z \in \mathbb{R}^2$ for Ant and Humanoid, and 16-dimensional discrete skills for other environments. In CSD, a 16-dimensional discrete skill vector is used for all environments. For both METRA and CSD, continuous skills are sampled from a standard Gaussian distribution and normalized to have unit norm, and discrete skills are designed to be zero-centered one-hot vectors. For DADS, we use 2-dimensional continuous skills sampled from the uniform range $[-1, 1]^2$ for Ant and Humanoid, and 16-dimensional one-hot vectors for the remaining environments. In DIAYN, we use 16-dimensional one-hot vectors across all environments. For training the low-level policy for downstream tasks, we use 16-dimensional discrete skills for all baseline methods.

**Adaptive sampling method**   For the adaptive sampling method, we evaluate the periodic skill policy conditioned on the current boundary periods every 1k episodes. To measure performance, we roll out 5 episodes and compute the average cumulative sum of $r_{\text{PSD}}$, defined as $\bar{R}_L = \mathbb{E}_{p(\tau | L)}[\sum_{t=0}^{T-1} r_{\text{PSD}}]$. For the threshold coefficients $\alpha$ and $\beta$, we found that setting $\alpha = 0.9$ and $\beta = 0.4$ works well in practice. To avoid abrupt narrowing of the radius range in the early stages of training, each bound is allowed to shrink only after it has first been expanded, i.e., after $\bar{R}_L > 0.9T$ has been satisfied at least once. The full algorithm is described in Algorithm 2, and a complete list of hyperparameters is provided in Table 2.

**Training PSD with pixel-based observations**   For experiments using pixel-based observations, we use a CNN-based encoder [51] to process visual inputs. To capture temporal continuity, we concatenate consecutive frames as input. We also apply random cropping as a form of data augmentation, following CURL [49]. We found that action repeat was not necessary to achieve stable training in our setup. A complete list of hyperparameters is provided in Table 3.

**Task-specific reward**   Since PSD is designed to enrich the agent's behavior with additional diversity while still achieving the primary task, we optionally combine the velocity-based external reward $r_{\text{ext}}$ with the intrinsic reward $r_{\text{PSD}}$.

Given that $r_{\text{PSD}} = \exp(-\kappa \Delta^2)$ is bounded in the range $(0, 1]$, we design the external reward to also have an upper bound of 1, ensuring a balanced contribution of both rewards when the agent reaches optimal performance, as follows:

$$r_{\text{ext}}(v_x) = \begin{cases} 1.0 & \text{if } v_x \geq v_x^* \\ v_x / v_x^* & \text{otherwise} \end{cases}$$

This reward function assigns $r_{\text{ext}} = 1.0$ when the agent's forward velocity $v_x$ exceeds the threshold $v_x^*$, and increases linearly as $v_x$ approaches the threshold from below. We set $v_x^* = 0.5$ for Ant and HalfCheetah, and $v_x^* = 1.0$ for Humanoid, Hopper, and Walker2D.

**Training the high-level policy for downstream tasks**   For downstream tasks, we train the high-level policy using PPO [80] while keeping the low-level skill policies frozen. This training utilizes a task-specific reward and an additional observation, $s_{\text{task}}$, which are detailed in Appendix C.1. In all experiments, the high-level policy, $\pi^h(L | s_{\text{task}}, s)$ (or $\pi^h(z | s_{\text{task}}, s)$ for baseline methods), selects a skill every $H$ environment steps, and the low-level policy then executes this skill for the subsequent $H$ steps. The high-level policy is trained for 100k episodes using the hyperparameters listed in Table 4.

**Training PSD combined with METRA**  METRA [67] learns an encoder $\phi_m$ and a skill policy $\pi(a \mid s, z)$ that encourages transitions to deviate maximally along latent directions $z$, while constraining the one-step latent distance to capture temporal coherence. As described in the original paper [67], the constrained objective in Eq. (9) is optimized by maximizing the following components:

$$\mathcal{J}_{\text{METRA},\phi_m} = \mathbb{E}_{(s,s',z)\sim\mathcal{D}} \left[ (\phi_m(s') - \phi_m(s))^\top z + \lambda_m \cdot \min\left(\epsilon, 1 - \|\phi_m(s') - \phi_m(s)\|_2^2\right) \right], \quad (11)$$

$$\mathcal{J}_{\text{METRA},\lambda_m} = -\lambda_m \cdot \mathbb{E}_{(s,s',z)\sim\mathcal{D}} \left[ \min\left(\epsilon, 1 - \|\phi_m(s') - \phi_m(s)\|_2^2\right) \right], \quad (12)$$

where $\lambda_m$ is a Lagrange multiplier, updated during training via the dual gradient method to enforce the constraint.

A naïve combination of PSD and METRA—training their encoders *independently* and simply summing their intrinsic rewards—fails in practice. As explained in Section 2, the METRA objective strongly favors skills with the shortest possible period. This is because shorter periods typically correspond to faster motions, which lead to larger per-step deviations in the latent space and thus yield higher values of $(\phi_m(s') - \phi_m(s))^\top z$. Consequently, all discovered skills collapse into a single short-periodic behavior, undermining the diversity of the learned skill of PSD.

To address this issue, we condition each encoder on the other method's skill variable by incorporating it as an additional input to the state. Specifically, we augment the input to $\phi_L$ with the skill variable $z$ from METRA, and the input to $\phi_m$ with the period variable $L$ from PSD, resulting in:

$$\phi_L(s) \longrightarrow \phi_L(s, z), \quad \phi_m(s) \longrightarrow \phi_m(s, L).$$

This mutual conditioning allows each encoder to account for the temporal properties imposed by the other method, thereby regularizing their joint optimization and preventing skill collapse. For example, from the perspective of training $\phi_m(s, L)$, the METRA objective in Eqs. (11) and (12) encourages latent representations that exhibit large per-step deviations in the latent space while satisfying the periodicity determined by $L$.

By jointly optimizing both encoders with this conditioning, we obtain a policy $\pi(a \mid s, z, L)$ that can independently modulate both the temporal *direction* (i.e., the skill variable $z$) and the temporal *length* (i.e., the period variable $L$) in a fully unsupervised manner. The full algorithm is described in Algorithm 3 and a complete list of hyperparameters is provided in Table 5.

**Latent Space Dimensionality**  As summarized in Table 2, we used $\{3, 6\}$-dimensional latent spaces across all embodiments, which we found to work well in practice. In contrast, a 2-dimensional latent space (i.e., a plane) led to unstable performance for complex agents such as Ant or Humanoid. We hypothesize that, since the PSD objective does not explicitly constrain the latent circles for each $L$ to lie on the same plane, having additional degrees of freedom allows different circles to occupy different planes. This, in turn, helps stabilize the embedding during training. Conversely, when the latent space is restricted to only 2 dimensions, this flexibility is lost, which leads to instability.

## C.3  Visualizations

**PCA visualization for latent space**  As described in Section 3.2 and Appendix C.2, the circular latent space of PSD is not necessarily 2-dimensional. Given the PSD formulation, we map states $s$ to latent vectors $\phi_L(s)$ with 3 or more dimensions in practice to better capture periodicity. To visualize this circular latent space, as shown in Figure 5 and our video, we apply Principal Component Analysis (PCA) to obtain a 2-dimensional projection that clearly illustrates the underlying circular structure.

## C.4 Full Algorithm of Adaptive Sampling Method

---

**Algorithm 2** Adaptive Sampling Method

---

1: **Initialize**: policy $\pi$, encoder $\phi$, current sampling bound $L_{\min}, L_{\max}$
2: *updated_once_min* $\leftarrow$ `False`
3: *updated_once_max* $\leftarrow$ `False`
   // Evaluate current bounds at $L_{\min}$ and $L_{\max}$
4: **for** each evaluation episode **do**
5:     **for** $L \in \{L_{\min}, L_{\max}\}$ **do**
6:         Execute $\pi(a \mid s, L)$ for the entire episode
7:         Compute cumulative reward $\sum_{t=0}^{T-1} r_{\text{PSD}}(L)$
8:     **end for**
9: **end for**
10: Compute average reward $\bar{R}_{L_{\min}}, \bar{R}_{L_{\max}}$
   // Update current bounds
11: **if** $\bar{R}_{L_{\min}} > \alpha T$ **then**
12:     $L_{\min} \leftarrow L_{\min} - N$
13:     *updated_once_min* $\leftarrow$ `True`
14: **end if**
15: **if** $\bar{R}_{L_{\max}} > \alpha T$ **then**
16:     $L_{\max} \leftarrow L_{\max} + N$
17:     *updated_once_max* $\leftarrow$ `True`
18: **end if**
19: **if** $\bar{R}_{L_{\min}} < \beta T$ **and** *updated_once_min* = `True` **then**
20:     $L_{\min} \leftarrow L_{\min} + N$
21: **end if**
22: **if** $\bar{R}_{L_{\max}} < \beta T$ **and** *updated_once_max* = `True` **then**
23:     $L_{\max} \leftarrow L_{\max} - N$
24: **end if**
25: **return** $L_{\min}, L_{\max}$

---

## C.5 Full Algorithm of PSD Combined with METRA

---

**Algorithm 3** PSD combined with METRA

---

1: **Initialize**: policy $\pi$, PSD encoder $\phi$, METRA encoder $\phi_m$, sampling bound $L_{\min,\max}$, replay buffer $\mathcal{D}$, Lagrange multiplier $\lambda_{1,2,m}$
2: **for** each training epoch **do**
3:     Update $L_{\min}, L_{\max}$ **if** *AdaptiveSampling* is enabled
   // Environment interaction
4:     **for** each episode in the epoch **do**
5:         Sample $L \sim p(L)$ where $L \in [L_{\min}, L_{\max}]$
6:         Sample $z \sim p(z)$ where $p(z)$ is the skill prior of METRA
7:         Execute $\pi(a \mid s, z, L)$ for the entire episode, and store transitions $(z, L, s_t, a_t, s_{t+1})$ in $\mathcal{D}$
8:     **end for**
   // Update encoders and RL network
9:     Update $\phi_L(s, z)$ by maximizing $\mathcal{J}_{\text{PSD},\phi}$ using samples $(z, L, s_t, s_{t+1})$ from $\mathcal{D}$
10:     Update $\phi_m(s, L)$ by maximizing $\mathcal{J}_{\text{METRA},\phi_m,\lambda_m}$ using samples $(z, L, s_t, s_{t+1}, s_{t+L})$ from $\mathcal{D}$
11:     Compute intrinsic reward $r_{\text{PSD}}$
12:     Compute intrinsic reward $r_{\text{METRA}}$
13:     Update $\pi(a \mid s, z, L)$ with $\alpha_{\text{PSD}} \cdot r_{\text{PSD}} + r_{\text{METRA}}$ using SAC
14: **end for**

---

## C.6 Hyperparameters

Table 2: Hyperparameters for training PSD.

| Parameter | Value |
|---|---|
| Learning rate | $1 \times 10^{-4}$ |
| Discount factor $\gamma$ | 0.99 |
| Optimizer | Adam [48] |
| $N$ of episodes per epoch | 8 |
| $N$ of gradient steps per epoch | 64 |
| Replay buffer size | $5 \times 10^5$ |
| Minibatch size | 1024 ($\phi_L$), 256 (others) |
| Target smoothing coefficient | 0.995 |
| Entropy coefficient | Auto-tuned |
| Circular latent dimension $d$ | $\{3, 6\}$ |
| Output dimension of the positional encoding $D$ | 8 |
| $r_{\mathrm{PSD}}$ $\kappa$ | 10 |
| $\mathcal{J}_{\mathrm{PSD}}$ $\epsilon$ | $10^{-5}$ |
| $\mathcal{J}_{\mathrm{PSD}}$ $k$ | 0.5 |
| $\mathcal{J}_{\mathrm{PSD}}$ $\lambda_1$ | 5 (Ant, HalfCheetah), 10 (Humanoid, Hopper, Walker2D) |
| $\mathcal{J}_{\mathrm{PSD}}$ $\lambda_2$ | 5 (Ant, HalfCheetah), 10 (Humanoid, Hopper, Walker2D) |
| $N$ of hidden layers | 2 |
| $N$ of hidden units per layer | 1024 |
| Step size of adaptive sampling $N$ | 1 |
| Adaptive sampling interval | 2000 episodes |
| $N$ of evaluation episodes for adaptive sampling | 5 |
| Thresholds $(\alpha, \beta)$ for adaptive sampling | $(0.9,\ 0.4)$ |

Table 3: Hyperparameters for training PSD with Pixel-based observation (others same as Table 2).

| Parameter | Value |
|---|---|
| Replay buffer size | $3 \times 10^4$ |
| Minibatch size | 512 ($\phi_L$), 256 (others) |
| Circular latent dimension $d$ | $\{3, 6\}$ |
| Output dimension of the positional encoding $D$ | 128 |
| $\mathcal{J}_{\mathrm{PSD}}$ $\lambda_1$ | 5 (Ant), 3 (HalfCheetah) |
| $\mathcal{J}_{\mathrm{PSD}}$ $\lambda_2$ | 5 (Ant), 3 (HalfCheetah) |
| Encoder | CNN [51] |
| Random crop | $90 \times 90 \times 3 \ \rightarrow \ 84 \times 84 \times 3$ |
| $N$ of stacked frames | 3 |
| $N$ of action repeat | 1 |

Table 4: Hyperparameters for training the high-level policy using PPO.

| Parameter | Value |
|-----------|-------|
| Learning rate | $3 \times 10^{-4}$ (actor), $1 \times 10^{-3}$ (critic) |
| Discount factor $\gamma$ | 0.99 |
| Optimizer | Adam [48] |
| Skill duration $H$ | 10 |
| $N$ of episodes per epoch | 4 |
| $N$ of gradient steps per epoch | 80 |
| Batch size | 256 |
| PPO clipping parameter $\epsilon$ | 0.2 |
| $N$ of hidden layers | 2 |
| $N$ of hidden units per layer | 256 |

Table 5: Hyperparameters for training PSD Combined with METRA.

| Parameter | Value |
|-----------|-------|
| Learning rate | $1 \times 10^{-4}$ |
| Discount factor $\gamma$ | 0.99 |
| Optimizer | Adam [48] |
| $N$ of episodes per epoch | 8 |
| $N$ of gradient steps per epoch | 64 |
| Reward coefficient $\alpha_{\text{PSD}}$ | 1.0 |
| Replay buffer size | $5 \times 10^{5}$ |
| Minibatch size | 1024 ($\phi_L$), 256 (others) |
| Target smoothing coefficient | 0.995 |
| Entropy coefficient | Auto-tuned |
| Skill dimension of METRA | 2-D cont. (Ant), 1-D cont. (Walker2D) |
| Circular latent dimension $d$ | 6 (Ant), 3 (Walker2D) |
| Output dimension of the positional encoding $D$ | 6 |
| $r_{\text{PSD}}$ $\kappa$ | 10 |
| $\mathcal{J}_{\text{PSD}}$ $\epsilon$ | $10^{-5}$ |
| $\mathcal{J}_{\text{PSD}}$ $k$ | 0.5 |
| $\mathcal{J}_{\text{PSD}}$ $\lambda_1$ | 5 (Ant), 10 (Walker2D) |
| $\mathcal{J}_{\text{PSD}}$ $\lambda_2$ | 5 (Ant), 10 (Walker2D) |
| $\mathcal{J}_{\text{METRA}}$ $\epsilon$ | $10^{-3}$ |
| $\mathcal{J}_{\text{METRA}}$ $\lambda_m$ | 30 |
| $N$ of hidden layers | 2 |
| $N$ of hidden units per layer | 512 (Ant), 1024 (Walker2D) |

# D  Additional Experimental Results

## D.1  Evolution of Learned Bounds via Adaptive Sampling Method

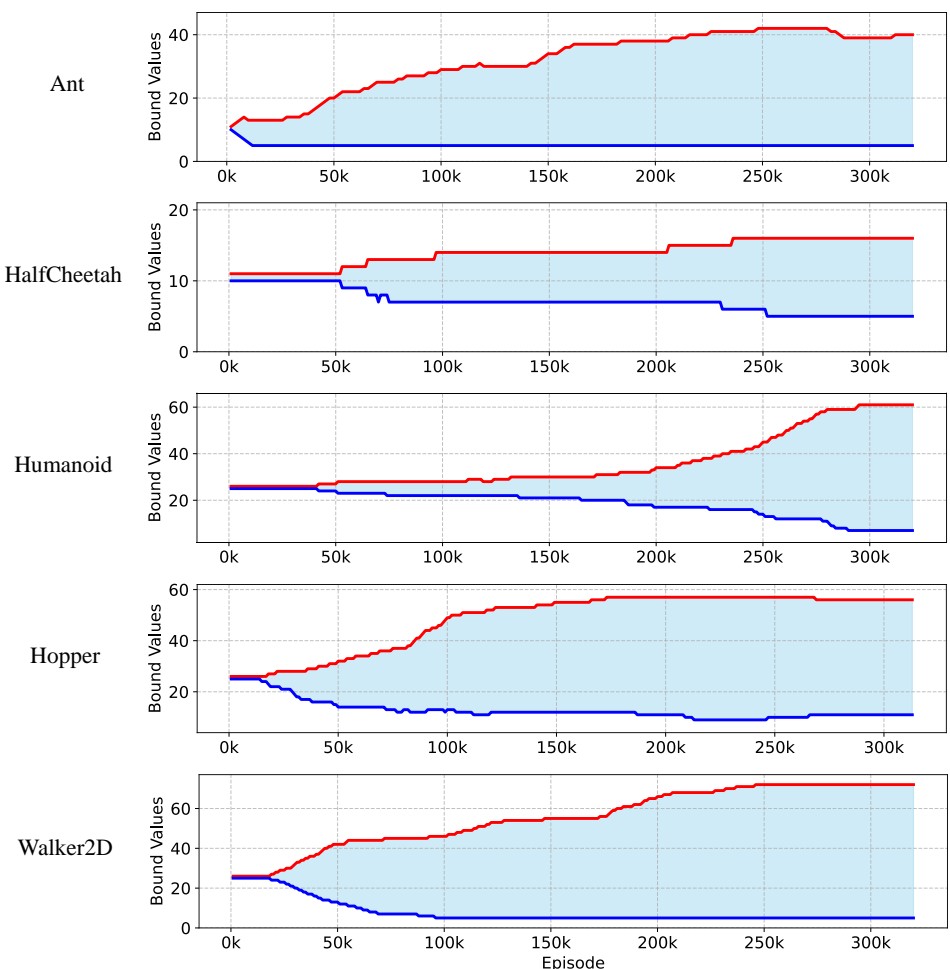

Figure 9: **Evolution of the $L_{\text{min}}$, $L_{\text{max}}$ during training.** The figure shows how the period variable $L_{\text{min}}$, $L_{\text{max}}$ evolves over training episodes with the adaptive sampling method applied to the Ant, HalfCheetah, Humanoid, Hopper, and Walker2D environments. As training progresses, increasingly challenging periods are proposed to the agent based on the average cumulative sum of $r_{\text{PSD}}$, enabling the discovery of a wider range of periodic behaviors.

In Figure 9, we visualize the evolution of the sampling range of $L$ during training with the adaptive sampling method. To prevent the period variable $L_{\text{min}}$, which must be a positive integer, from becoming too small, we set the minimum value $L_{\text{min}} = 5$.

Although training begins with a single period value, the adaptive sampling method gradually proposes more challenging periods, enabling the agent to acquire skills across a broad range of dynamically feasible period lengths. Moreover, since training is conducted with the combined reward of $r_{\text{ext}}$ and $r_{\text{PSD}}$ (as described in Appendix C.2), the agent learns to maintain a velocity above the target $v_x^*$ while acquiring maximally diverse skills to optimize the overall reward. A notable property of this method is that even if a proposed period is initially rejected due to low performance, the agent may later learn to handle it as training progresses. Overall, this method enables PSD to autonomously discover a wide range of periodic behaviors without requiring prior knowledge of agent-specific period scales.

## D.2 Skill Interpolation

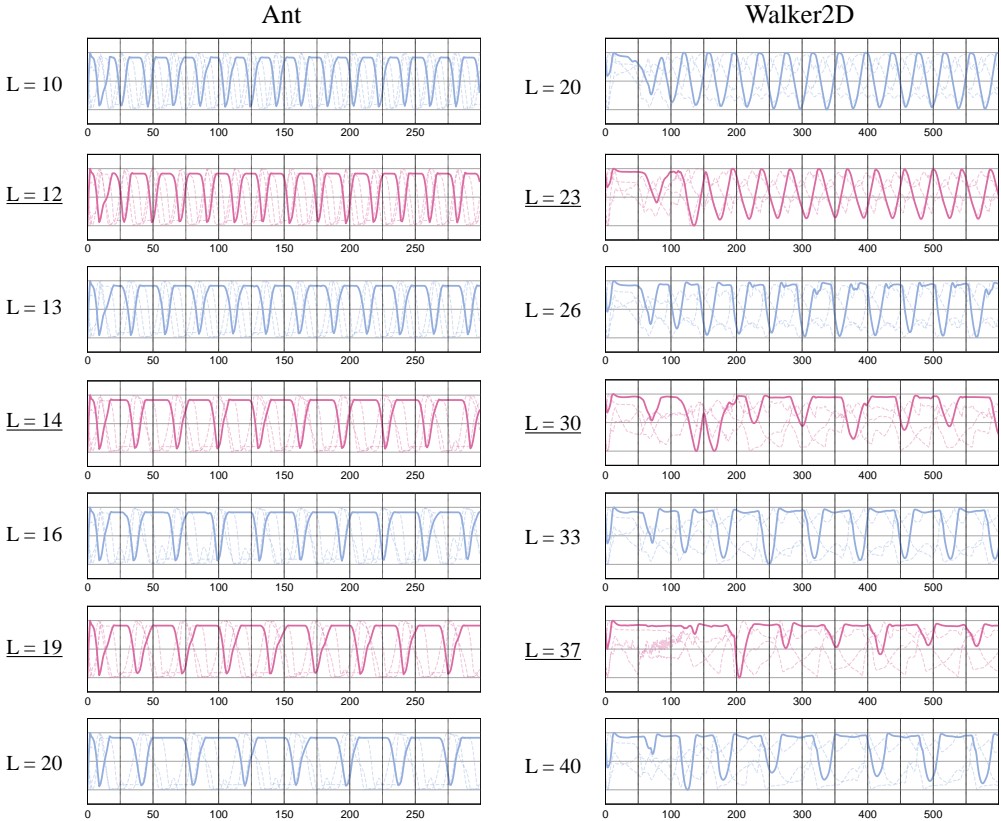

Figure 10: **Trajectories of the skill policy** $\pi(a \mid s, L)$ **under different values of** $L$**.** The figure shows representative joint trajectories of Ant (*left*) and Walker2D (*right*) generated by the skill policy $\pi(a \mid s, L)$ under different values of $L$. The blue trajectories are rollouts of the policy conditioned on the final sampling candidates after convergence, while the magenta ones are generated using intermediate integer values between these candidates. Although the magenta trajectory does not perfectly satisfy the $2L$ periodicity, it still generalizes well, indicating that our sampling strategy is effective and the circular representation of PSD generalizes across diverse periods.

## D.3 Additional Analysis of Pixel-based Observation Experiments

Comparing Figure 3 and Figure 7, we observe that the pixel-based HalfCheetah exhibits narrower and higher-frequency periodic behaviors than its state-based counterpart, despite having identical robot dynamics. We hypothesize that this is due to the inability to differentiate periodic variations in the vertical direction from pixel observations. As shown in our video, it is difficult to perceive $z$-axis variations from raw images, and this issue is exacerbated by random cropping. In contrast, in state-based settings, the $z$-coordinate is explicitly provided as the first dimension of the observation vector, making it easier for the neural network to recognize vertical changes. This suggests that the ability to represent vertical height enables PSD to learn longer-period behaviors (e.g., jumping) in the state-based setting, as is also visually demonstrated in the video. On the other hand, in the Ant environment, both pixel-based and state-based observations provide similar levels of information, and thus result in similar walking behaviors.

## D.4 Full Experimental Results of Figure 5

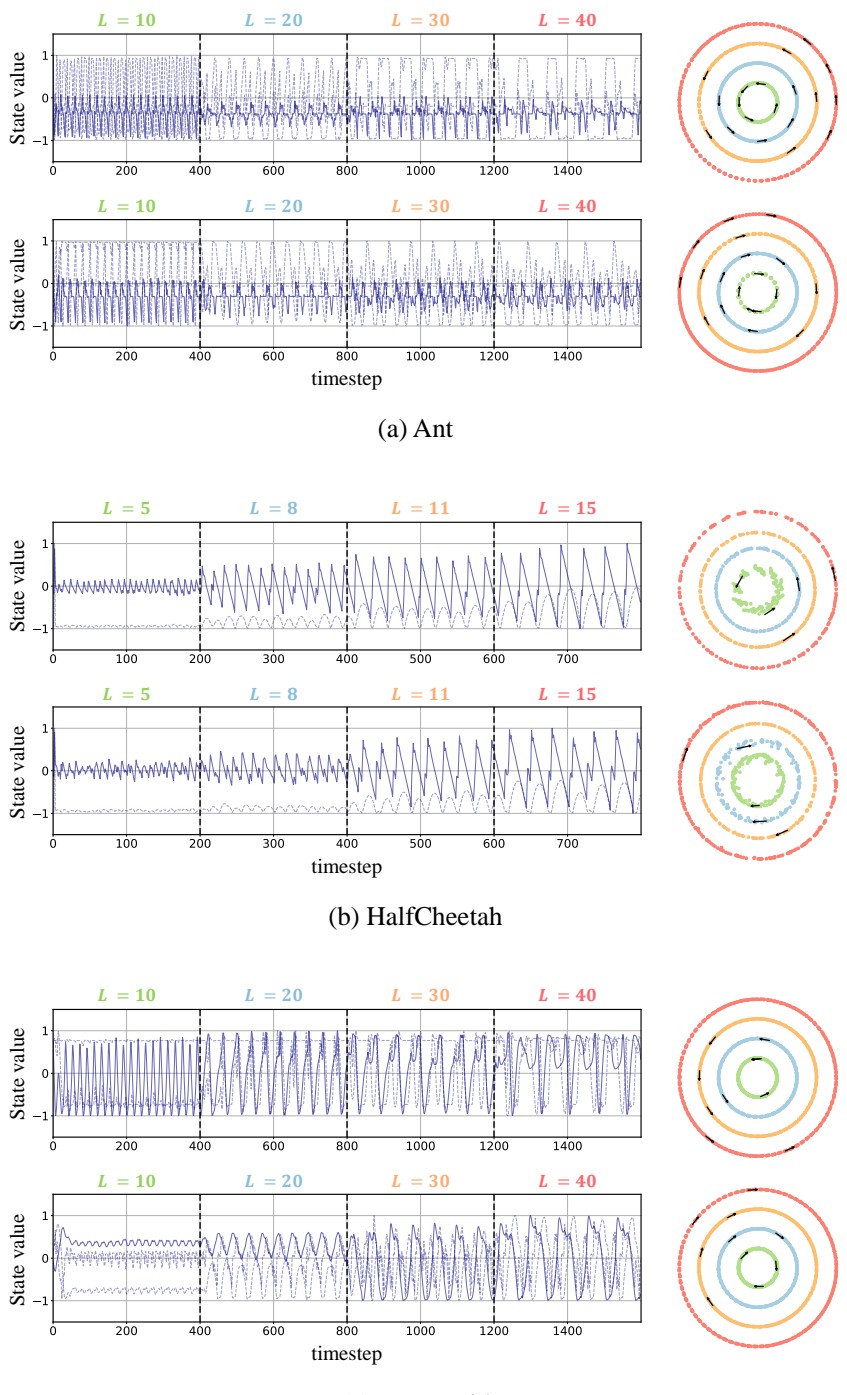

(a) Ant

(b) HalfCheetah

(c) Humanoid

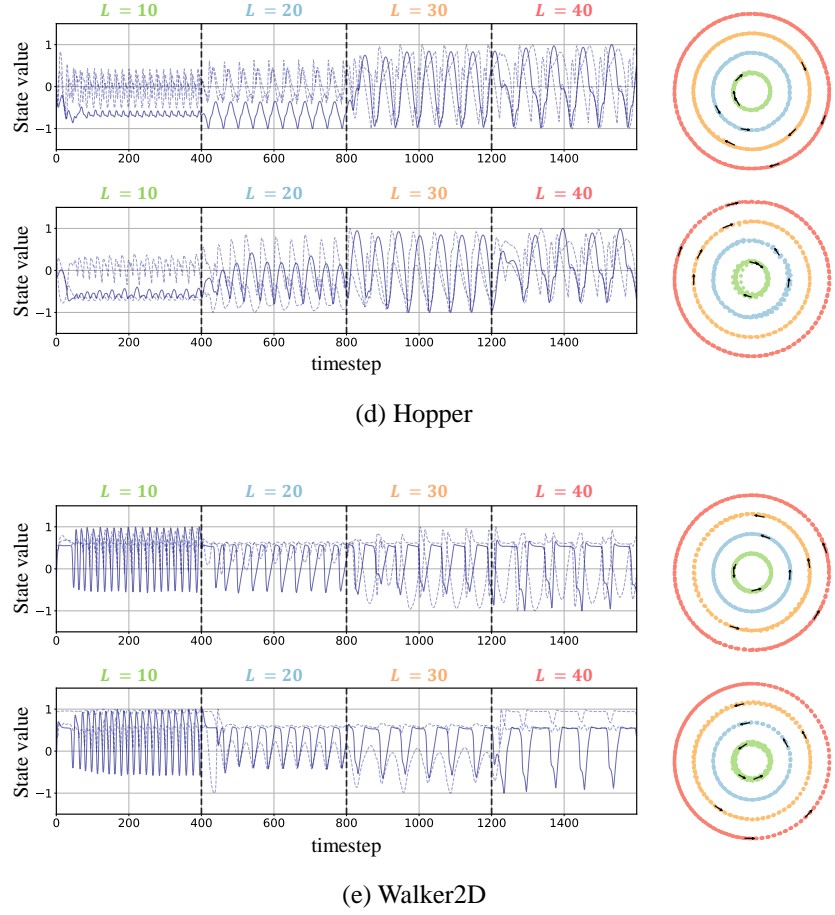

(d) Hopper

(e) Walker2D

Figure 11: **Full experimental results of Figure 5.** The figure shows the state trajectories and the 2D PCA projection of their latent encodings learned by PSD. Within a single episode, we *switch* the period variable $L$ at fixed time intervals.

## D.5 Full Experimental Results of Figure 6

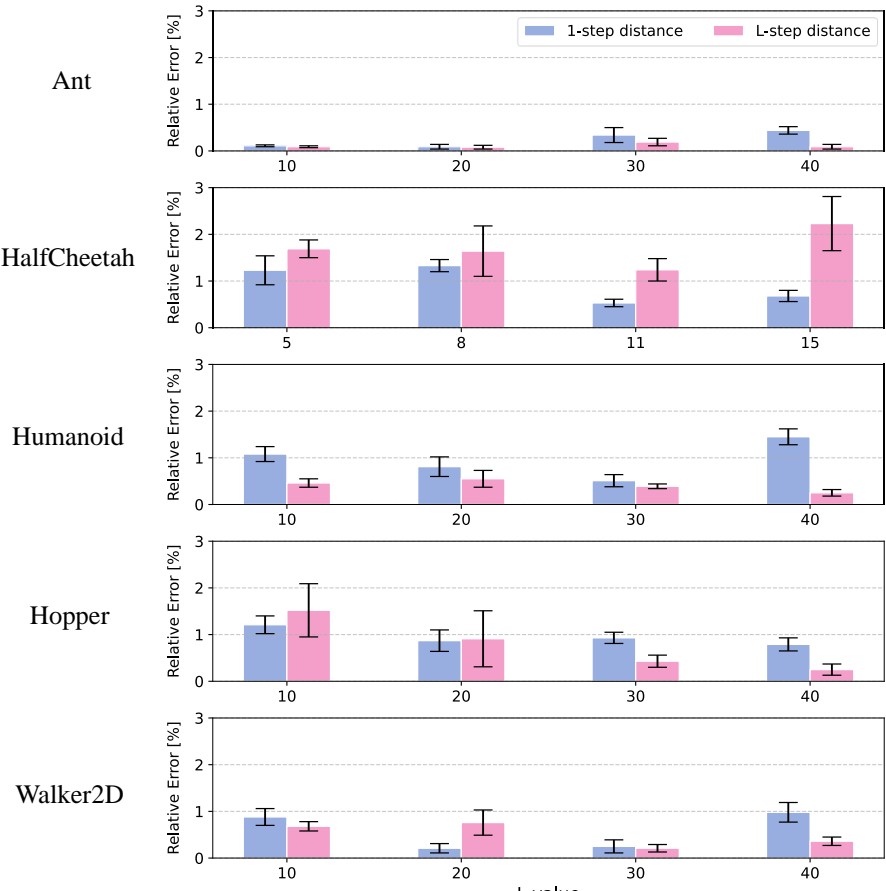

Figure 12: **Full experimental results of Figure 6 (4 seeds).** We compute the relative error of the learned temporal distance as $\text{Rel. Error} = \left| \frac{\text{Empirical average} - \text{Optimal value}}{\text{Optimal value}} \right| \times 100 \, [\%]$.

