# OpenReview forum: "Periodic Skill Discovery"
_NeurIPS.cc/2025/Conference — NeurIPS 2025 poster_

### Official Review · Reviewer_qG2Z · 2025-06-29

**Clarity:** 4
**Significance:** 2
**Originality:** 3
**Rating:** 5
**Confidence:** 3

**Summary:**

This work introduces a novel framework for unsupervised discovery of explicitly periodic skills in locomotion tasks called periodic skill discovery (PSD). This is done by learning a latent space in which, when embedded, trajectories are incentivised to follow a circular path with some known, fixed period. Conversely, an intrinsic reward is obtained from this latent embedding in which a period-conditioned policy is optimised with reinforcement learning to maximise the fit of the trajectory to a circle inside the latent space. These two approaches bootstrap off each other to result in the unsupervised discovery of periodic policies. The results section shows that, compared to prior unsupervised skill discovery work, PSD discovers locomotion skills with a greater diversity of periods.

**Questions:**

Main question (state space)
- I think my most fundamental lack of understanding comes from the definition of what a state is. Presumably, the state used for the latent embedding does not include the global positional co-ordinates of the body? Otherwise a policy would presumably only be periodic if it returned to the same position. Is the state then made up of only the relative positions/rotations/velocities of the constituent parts of the body?
- This feels like a fundamental limitation of the approach - can it still capture periodicity of a trajectory if only a subset of the state is periodic? e.g. if the global position is included in the state

Other minor points / questions
- The problem statement (online unsupervised skill discovery) could be more clearly outlined in the introduction. If one was not familiar with the prior work in this area it is unclear exactly what the high level problem is and why this is useful.
- Why is only the 'one-step' periodic constraint (corresponding to Eq. 6) used as the intrinsic reward while the 'diameter' constraint (corresponding to Eq. 5) is not?
- Line 182 "Since we want to discover maximally diverse period" is either bad grammar or confusing at least
- My understanding of section 3.4 is that the period limits essentially dynamically expand according to whether the policies on the limits are satisfying the periodic conditions. Is their a practical maximum here? It would seem like you could come up with policies of essentially unboundedly high period and with low periods of a single timestep (e.g. staying still).
- This approach feels quite related to quality diversity methods (e.g. MAP elites [1]) where $L$ corresponds to a behavioural descriptor. Especially in regards to the main result alluding to how the diversity of periods is a useful thing. This link should be discussed.

[1] - Jean-Baptiste Mouret, Jeff Clune. Illuminating search spaces by mapping elites. 2015.

**Ethical Concerns:**

["NO or VERY MINOR ethics concerns only"]

**Final Justification:**

The authors response suitably clarified all the questions I raised.
As outlined in my original review and response, I think the most significant weakness of this method is how reliant it is on the given observation space to be entirely periodic - I do not think the method as it stands would generalise to pixel-based environments with arbitrary (i.e. non-repeating) terrain for instance. That being said, I do feel like the method as it stands is significant enough to warrant acceptance.

**Limitations:**

yes

**Quality:**

3

**Strengths And Weaknesses:**

Strengths
 - The formulation of the periodic objective is intuitive, well explained and original
 - The paper is well presented and easy to read

Weaknesses
 - I would consider the significance of the paper to be a practical inductive bias, however like all inductive biases it must be questioned if this is actually a useful thing or will eventually be surpassed by simply increasing the data, scale and compute of a more general method. Specifically, a generic unsupervised skill discovery method *should* be able to discover different gaits of different periods if appropriately conditioned on full trajectories of full state representations.
- It seems like the method could not be applied to an offline dataset of trajectories, as the target period $L$ of a trajectory must be known for learning the latent space, limiting the applicability of the method to online approaches.
- As far as I can tell the current setup cannot represent different gaits of the same period

For these reasons I am giving a borderline accept, but would be open to increasing my score - see questions section

---

> ### Author Rebuttal · Authors · 2025-07-31
>
> Dear Reviewer qG2Z,
>
> We sincerely thank you for your detailed and insightful feedback.
> We have prepared our responses below:
>
> ---
> **Q1. The paper introduces a practical inductive bias. Will it be eventually be surpassed by simply increasing the data, scale and compute of a more general method?**
>
> **A1.**
> We appreciate your comment. Even as generalist methods grow in data, scale, and compute, an explicit inductive bias for *periodicity* will continue to fill a role that pure scale alone cannot guarantee—much like convolution remains valuable in vision despite ever-larger transformers.
>
> At the same time, this inductive bias is not at odds with future progress in large-scale models; it complements it. Since one of the PSD’s contribution is to discover a maximally diverse repertoire of useful periodic behavior in an unsupervised manner, it can be plugged in as a lightweight sub-module that a large, general model calls whenever rhythmic style or behavior-style rewards should be injected. In this way, the large model continues to handle broad task understanding and high-level decisions, while PSD provides a principled, sample-efficient mechanism for adding rhythmic nuance, thereby enhancing versatility rather than competing with scale.
>
> **Q2. Applicability of PSD to offline datasets.**
>
> **A2.** Thank you for your insightful comment regarding the applicability of PSD to offline datasets.
>
> The main prerequisite is to know or estimate the period variable $L$ for each trajectory in the offline dataset, which we believe is a tractable problem.
>
> For instance, we could estimate $L$ for a given trajectory using a Fourier transform to find its dominant frequency, similar to our analysis in Figure 3. Alternatively, one could add an auxiliary reconstruction loss to the existing PSD objective to find the period $L$ that best represents a trajectory's periodic features.
>
> Once the period $L$ for each offline trajectory is estimated, an offline skill learning process is available. The recipe would be as follows:
>
> 1. For each trajectory in the offline dataset, first estimate its period, $L$.
> 2. Use these estimated (trajectory, L) pairs to train the PSD encoder and its circular latent space.
> 3. Finally, with the learned representations, train an RL policy with $r_{psd}$ using a standard offline RL algorithm.
>
> Additionally, an offline-to-online approach could allow the agent to discover skills with periods that **go beyond the range** covered by the original dataset, while remaining consistent with the behaviors it contains. Thank you again for the valuable suggestion.
>
> **Q3. As far as I can tell the current setup cannot represent different gaits of the same period.**
>
> **A3.** We appreciate your excellent comment. The primary contribution of this work lies in autonomously discovering a diverse range of *periods*, rather than directly targeting multimodality within a single period.
>
> However, your comment raises a very interesting direction for future work. Inspired by your idea, encoding the representation onto a more general geometry, such as the surface of a sphere (instead of a circle), would allow different gaits to be represented even for a single period. For example, different gaits of the same period could be encoded as distinct circles of the same circumference on the sphere's surface. We agree this is a very exciting future direction to explore.
>
> **Q4. Question about the state space of PSD.**
>
> **A4-1.** Your presumptions are correct. The default state representation for PSD does not include the global XY base position and consists only of proprioceptive states like joint angles and heading angles, as described in Appendix B. This corresponds to the default observation space in Gym environments and is a common practice in prior work [1, 2, 3] on robot learning for generating diverse behaviors.
>
> **A4-2.** When we trained the Ant agent with the full state (including the XY coordinates) using only the PSD objective, we observed that the agent vigorously wiggled in place, making body movements in sync with the period. This behavior is the result of the optimization satisfying our objective, as it makes both the joint angles and the global XY coordinates periodic.
>
> **A4-3.** As stated in our previous response, PSD can discover periodic behaviors even when the state includes the XY coordinates. However, a conflict arises when the agent must also satisfy a specific downstream task. If an external task reward encourages non-periodic changes in the XY position (e.g., by rewarding forward velocity), this would directly conflict with the PSD objective, which requires the latent representation to be identical after $2L$ steps, and would therefore hinder the learning process.
>
> [1] Peng, Xue Bin, et al. "Amp: Adversarial motion priors for stylized physics-based character control." *ACM Transactions on Graphics (ToG)* 40.4 (2021): 1-20.
>
> [2] Peng, Xue Bin, et al. "Ase: Large-scale reusable adversarial skill embeddings for physically simulated characters." *ACM Transactions On Graphics (ToG)* 41.4 (2022): 1-17.
>
> [3] Li, Chenhao, et al. "FLD: Fourier Latent Dynamics for Structured Motion Representation and Learning." *The Twelfth International Conference on Learning Representations*.
>
> **Q5. Need a clear outline of the problem statement in the introduction for online unsupervised skill discovery.**
>
> **A5.** Thank you for your suggestion. We will add an explanation of the basic setup of unsupervised skill discovery, along with the problem it aims to address and its significance, in the second paragraph of the introduction.
>
> **Q6. Use of the one-step periodic constraint as the intrinsic reward instead of the diameter constraint.**
>
> **A6.** That is an excellent question regarding our reward design! We chose the one-step periodic constraint instead of the diameter constraint for two primary reasons: learning efficiency and compatibility with off-policy RL algorithms.
>
> **A6-1.** On a practical level, the one-step reward provides a dense, per-timestep learning signal, which is significantly more effective for policy training than the sparse reward from the diameter constraint that is only evaluated over an $L$-step horizon.
>
> **A6-2.** More fundamentally, the use of a single-step reward is crucial for general compatibility with standard off-policy algorithms (e.g., SAC, DDPG). In contrast, a reward based on the diameter constraint would be inherently a multi-step, sparse reward, as it depends on states that are $L$ timesteps apart. This choice was made to ensure that PSD is easy to use and broadly compatible with standard off-policy RL algorithms that learn from single-step transition tuples. However, as you suggest, a more advanced implementation could certainly be developed to handle the multi-step diameter constraint, perhaps by leveraging n-step returns, which is an interesting direction for future work.
>
> Thus, we chose this approach to ensure a dense learning signal and broad applicability with standard off-policy RL algorithms.
>
> **Q7. Line 182 "Since we want to discover maximally diverse period" is either bad grammar or confusing at least.**
>
> Thank you for your valuable suggestion. We found that the original wording was inconsistent with the tone of the paper.
>
> We have revised the sentence as follows: "**To enable the agent to discover a maximally diverse range of periods** without any prior knowledge of the agent-specific period ranges, we introduce an adaptive sampling method that …”
>
> **Q8. Is their a practical maximum of discovered period? It would seem like you could come up with policies of essentially unboundedly high period and with low periods of a single timestep (e.g. staying still).**
>
> **A8-1.** Thank you for the question. Your understanding is correct. the period’s upper & lower bounds dynamically expand based on whether the policies at the boundaries satisfy the periodic conditions.
>
> **A8-2.** Regarding the practical maximum, while there is no hard-coded upper bound, the range is limited by what is **dynamically feasible** for the agent, especially when maintaining performance on a specific task. For example, there is likely a maximum period at which an agent can still maintain a forward velocity of 2m/s.
>
> **A8-3.** For the lower bound, since $L$ must be a positive integer in our objective, the shortest possible period corresponds to $L=1$, resulting in a period of 2 timesteps.
>
> **Q9. Connection to quality diversity methods such as MAP-Elites.**
>
> Thank you for this excellent suggestion. You are correct that our approach has a link to quality diversity (QD) methods such as MAP-Elites. However, we would like to highlight a key difference. While this QD methods typically operate on a pre-defined, static grid, PSD dynamically adjusts the range of the descriptor ($L$) based on the agent’s performance (sum of reward) and the embodiment's dynamics.
>
> Following your suggestion, we will further investigate this connection and add a discussion of both the similarities and this key distinction to our Extended Related Work section.
>
> ---
>
> Thank you again for your valuable and insightful review. Also, please let us know if our responses have addressed your questions. If anything requires further clarification, please do not hesitate to let us know.

---

> ### Comment · Reviewer_qG2Z · 2025-08-04
>
> Thank you for the detailed response to my review.
>
> Q1 - satisfactory answer
>
> Q2 - This is a fair answer - it would be nice to see this as an experiment but I don't think necessary for acceptance
>
> Q3 - Again this would be nice to see but not necessary for acceptance
>
> Q4 - I think this is still my biggest gripe with the method. Rather than being simply plug-and-play, this does severely limit the applicability of this method to apply to observation spaces where we know a priori which features we wish to be periodic. In the symbolic space this is not too large an issue as you can simply mask out the non-periodic features, however this is not possible in e.g. pixel-based environments. Your results with pixel-based observations would seem to refute this, however my intuition is that the only reason these worked is because the rendering of the floor is periodic (the checkerboard pattern). You could imagine that in navigating a non-periodic environment e.g. a landscape with variable terrain, then PSD with pixel observations would be forced to return to its original position every cycle to satisfy the periodic constraint (as with the wiggling ant). It would be interesting to try something like co-learning a projection in the latent space to a lower dimensional space, and then only forcing the periodicity to apply in this lower dimensional space. The hope would then be that the features we didn't wish to be periodic (e.g. XY coords) could be 'projected out' and ignored.
>
> Having said all this, I think the novelty of the method in this paper is still sufficient, but this could be an interesting route forward to make this method more generally applicable.
>
> Q5 - thank you for adding this
>
> Q6 - This makes sense and is a fair justification, it would be nice to have it justified in the paper
>
> Q7 - Thanks I think that reads better
>
> Q8 - Thank you for clarifying, I can see now that with the forward velocity constraint this makes more sense
>
> Q9 - Thank you for adding this comparison.
>
> Overall I am satisfied with the authors response. I think that more could be done with this method to increase its usability, especially wrt cases where only part of the latent space is periodic, however I think the paper is a significant enough contribution already. I will increase my score to a 5 and advocate for the acceptance of this paper.

---

> > ### Author Response · Authors · 2025-08-05
> > **Thank you for your response**
> >
> > Dear Reviewer qG2Z,
> >
> > Thank you very much for raising your score to 5.
> >
> > We are glad our answers addressed your questions and sincerely appreciate all the valuable suggestions you've provided.

---

### Official Review · Reviewer_N2n9 · 2025-07-02

**Clarity:** 3
**Significance:** 2
**Originality:** 4
**Rating:** 4
**Confidence:** 4

**Summary:**

The authors observe that locomotion skills often require periodic motion patterns which is largely overlooked in unsupervised skill discovery methods. To address this, they propose Period Skill Discovery (PSD), a framework for discovering periodic behaviors in an unsupervised way. Given a randomly sampled duration variable $L$, PSD trains an encoder to map states into a circular latent space with periodicity $2L$.  An exploration policy is trained to move along the circular space in a predetermined frequency depending on $L$. The method is evaluated on a set of locomotion tasks using either state-based or pixel-based inputs. The authors qualitatively demonstrate that the learned behaviors exhibit distinct periodic patterns. Additionally, they show that these skills can support downstream task performance when combined with extrinsic rewards.

**Questions:**

1. Frequency domain analysis (Fig. 3): Could you provide an intuitive explanation for why a Fourier transform of PCA-projected, normalized skill trajectories is informative? What does this metric reveal about the learned behaviors?
2. Robustness: Figures 3, 5, and 6 appear to be generated from a single random seed. Could you repeat these experiments across multiple seeds to assess the robustness of PSD? Similarly, Figure 6 presents histograms for just two $L$ values. How do these results vary across the full range of periodicities?
3. Latent space dimensionality: How is the dimensionality of the latent space $S$ selected? How does it affect the resulting behaviors? Also, since the 2D visualizations are PCA projections, it would be helpful to note this directly in the figure captions.
4. Sampling of $L$: The step-size adaptation for sampling L introduces several hyperparameters ($N$, $\alpha$, $\beta$) and lacks clear justification. Did the authors consider simpler alternatives? Can they comment on the impact of these hyperparameters?
5. Pixel-based task setup: In pixel-based settings, robot limbs are differently colored. Was this necessary for the method to work? An ablation or explanation would clarify its impact.
6. PSD + METRA setup: The combination with METRA is described as “fully unsupervised.” What aspect of vanilla PSD is supervised? Could METRA-PSD also be enhanced with extrinsic rewards for improved downstream performance?
7. Method scope: Is PSD applicable beyond locomotion domains? How might it perform (or be adapted) for tasks like object manipulation? Experiments in non-locomotion domains could clarify the method’s broader applicability.

**Ethical Concerns:**

["NO or VERY MINOR ethics concerns only"]

**Final Justification:**

The paper presents a solid contribution with a clearly written presentation, effective visualizations, and an original core idea. The proposed method is evaluated against reasonable baselines, and several of my initial concerns (e.g., step-size adaptation and missing ablations) were addressed through the explanations and additional experiments provided in the rebuttal.
However, due to the method’s specific focus on locomotion tasks, I would have appreciated a more general quantitative assessment of the discovered skills beyond frequency-domain analysis. In particular, how variable the skill discovery is across seeds and how reliably useful skills are discovered remains somewhat unclear (especially for embodiments not tested on downstream tasks). This limits the understanding of the method’s applicability to new domains.
Overall, while some open questions remain, I find that the reasons to accept the paper outweigh these concerns.

**Limitations:**

The limitations are only partially addressed. The paper includes a subsection on limitations, but it leans more toward outlining future work than identifying concrete shortcomings.

**Quality:**

2

**Strengths And Weaknesses:**

Strengths:
- clear presentation, especially in the method section. Equations are well-integrated with intuitive descriptions.
- strong qualitative visualizations that help illustrate the method and its motivation.
- the use of circular latent states to encourage periodicity is well-motivated and original.
- the method is evaluated against reasonable baselines from the unsupervised skill discovery literature.
- the related work section is thorough and well-positioned.

Weaknesses
- the discrete step size adaptation of $L$ feels somewhat ad hoc and lacks theoretical justification.
- behavioral analysis feels anecdotal. The experiments seem to be based on single-seed visualizations. A more systematic quantitative analysis across multiple seeds would strengthen the evaluation.
- the focus on locomotion tasks may limit the method’s generality and significance.
- missing ablations: Design choices, such as the dimensionality of the latent space $S$, the hyperparameters for sampling periodicity $L$, and the use of colored robot limbs (see questions below), are not systematically ablated or justified. Combined with the lack of discussion around where PSD may fail or underperform, this makes it difficult to assess the method’s true limitations and general applicability.

---

> ### Author Rebuttal · Authors · 2025-07-31
>
> Dear Reviewer N2n9,
>
> We are grateful for your careful reading and valuable suggestions.
>
> We have summarized the weaknesses and questions you raised and provide our responses below:
>
> ---
>
> **Q1. The discrete step size adaptation of $L$ feels somewhat ad hoc and lacks theoretical justification.**
>
> **A1.** In the standard MDP framework, time progresses in discrete steps. Consequently, $L$ in the PSD objective is intended to encode timestep information. Therefore, it should be a positive integer. To discover maximally diverse periods — while respecting the fact that the minimum unit of time is one step — we set the default step size of the adaptive sampling method to its minimal feasible value, $N=1$.
>
> **Q2. A more systematic quantitative analysis across multiple seeds would strengthen the evaluation. Could you repeat the experiments for Figures 3, 5, and 6 across multiple seeds to assess the robustness of PSD?**
>
> **A2.** Following the reviewer’s suggestion, we have conducted additional experiments to address this.
>
> **A2-1.** (Figure 3) To quantitatively compare the diversity of learned skills, we measured the frequency width — defined as the difference between the maximum and minimum frequencies of the skill trajectories (i.e., the gap between the highest and lowest frequency of bar in Figure 3) — and averaged the results over 3 seeds. A larger value for this metric indicates a broader range of periodic behaviors.
>
> |  | **Ant** | **HalfCheetah** | **Humanoid** | **Hopper** | **Walker2D** |
> | --- | --- | --- | --- | --- | --- |
> | **PSD (Ours)** | **9.1 ± 0.1** | **6.9 ± 0.2** | **6.1 ± 0.4** | **7.3 ± 1.6** | **8.7 ± 0.3** |
> | **METRA** | 2.1 ± 0.4 | 4.2 ± 0.2 | 2.0 ± 0.7 | 3.7 ± 0.5 | 3.5 ± 1.3 |
> | **CSD** | 2.3 ± 0.2 | 2.7 ± 0.3 | 1.7 ± 0.2 | 1.6 ± 0.4 | 4.2 ± 0.7 |
>
> As the results show, PSD consistently discovers a much wider range of periodic behaviors compared to the baselines across all embodiments. The frequency width, which measures the diversity of the discovered periods, is significantly larger for PSD in every case.
>
> We excluded DADS and DIAYN from this analysis as their policies collapse to a nearly static "yoga pose"—moving slightly at the beginning of an episode and then remaining stationary—which results in a frequency close to zero.
>
> **A2-2 & A2-3** (Figures 5 & 6). As you suggested, we have run the experiments for Figures 5 and 6 across multiple seeds to assess their robustness. While the NeurIPS rebuttal policy prevents us from including the new plots, we can summarize our findings.
>
> The circular latent space structure shown in Figure 5 is learned robustly across all seeds. Interestingly, we often observe that different random seeds result in qualitatively different gaits for the same period, suggesting our method can find multiple diverse solutions. Our analysis for Figure 6 also confirms that the latent representation accurately reflects temporal distance, suggesting the learned distance converges to its optimal value.
>
> Due to the time/character limits of the rebuttal period, we will provide the detailed quantitative results for a wider range of $L$ values during the upcoming author-reviewer discussion period. Also, the full multi-seed visualizations for these figures will be included in the final version of our paper.
>
> **Q3. The focus on locomotion tasks may limit the method’s generality and significance.**
>
> **A3.** PSD objective is theoretically applicable to any domain that involves periodic components, as it does not require domain-specific knowledge for optimization. For instance, in manipulation settings, it can be applied to tasks such as rotating a cube with fingers, turning a valve wheel, or mixing a salad bowl. Moreover, given a set of low-level policies $\pi_{low}(a|s,z)$, where $z$ represents a low-level skill such as ‘pick’, ‘move’, or ‘place’, a high-level policy $\pi_{high}(z|s,L)$ can be trained with the PSD objective to select skills $z$ as actions for composing complex, looped behaviors. We believe this could be an interesting future extension of PSD.
>
> **Q4. Lacks clear justification for hyperparameters of adaptive sampling method.**
>
> **A4.** To justify our hyperparameter choices for the adaptive sampling method, we conducted an ablation study (averaged over 3 seeds) on the Ant environment. The results show a clear trade-off between the final range of discovered periods and the stability of the expansion process.
>
> | $(\alpha, \beta, N) = (0.4, 0.9, 1)$ | **episode = 0** | **episode = 100k** | **episode = 200k** | **episode = 300k** | **number of rollbacks** |
> | --- | --- | --- | --- | --- | --- |
> | upper bound | 20 ± 0 | 37.3 ± 6.1 | 41.6 ± 8.2 | 47.3 ± 4.1 | 2 ± 0.8 |
> | lower bound | 20 ± 0 | 5 ± 0 | 5 ± 0 | 5 ± 0 | 0 ± 0 |
>
> | $(\alpha, \beta, N) = (0.4, 0.9, 3)$ | **episode = 0** | **episode = 100k** | **episode = 200k** | **episode = 300k** | **number of rollbacks** |
> | --- | --- | --- | --- | --- | --- |
> | upper bound | 20 ± 0 | 40 ± 3.7 | 44 ± 2.4 | 47 ± 4.9 | 7.3 ± 3.2 |
> | lower bound | 20 ± 0 | 5 ± 0 | 5 ± 0 | 5 ± 0 | 0 ± 0 |
>
> | $(\alpha, \beta, N) = (0.5, 0.8, 1)$ | **episode = 0** | **episode = 100k** | **episode = 200k** | **episode = 300k** | **number of rollbacks** |
> | --- | --- | --- | --- | --- | --- |
> | upper bound | 20 ± 0 | 40.6 ± 5.3 | 47.3 ± 1.2 | 48.7 ± 3.2 | 17.3 ± 1.9 |
> | lower bound | 20 ± 0 | 5 ± 0 | 5 ± 0 | 5 ± 0 | 0 ± 0 |
>
> | $(\alpha, \beta, N) = (0.5, 0.8, 3)$ | **episode = 0** | **episode = 100k** | **episode = 200k** | **episode = 300k** | **number of rollbacks** |
> | --- | --- | --- | --- | --- | --- |
> | upper bound | 20 ± 0 | 42 ± 6.2 | 52 ± 5.1 | 42 ± 7.8 | 28.7 ± 4.8 |
> | lower bound | 20 ± 0 | 5 ± 0 | 5 ± 0 | 5 ± 0 | 0 ± 0 |
>
> For instance, the less conservative threshold set ($\alpha=0.5, \beta=0.8$) with a larger step size ($N=3$) initially discovers the widest range. However, this aggressive search proves unstable, as shown by the high number of "range rollbacks" (28.7). This instability eventually causes the discovered range to collapse later in training (from 52 at 200k episodes to 42 at 300k). A high number of rollbacks generally indicates a potential for instability in long-term training.
>
> In contrast, the setting ($\alpha=0.4, \beta=0.9, N=1)$ expands the range to a comparable margin (47.3) with minimal rollbacks (2.0), demonstrating a much more stable and efficient learning process. We checked that this parameter set performs well across all embodiments, and therefore selected it for providing the best balance between discovering a diverse skill repertoire and maintaining training stability.
>
> **Q5. Could you provide an intuitive explanation for why a Fourier transform of PCA-projected, normalized skill trajectories is informative? What does this metric reveal about the learned behaviors?**
>
> **A5.** For Figure 3, we normalized the skill trajectories for each dimension using statistics computed from random action rollouts. This was done to account for the different scales of the state dimensions and to visualize the periodicity of each dimension fairly. In addition, for the 1D FFT, we applied PCA to reduce the trajectories to a single dimension and visualized them in the frequency domain. This analysis revealed that, compared to the baselines, PSD exhibits behaviors spanning a wider range of frequencies, as indicated by the broader distribution of bars in the frequency domain chart.
>
> **Q6. How is the latent space dimensionality chosen and how does it affect the resulting behaviors?**
>
> **A6-1.** As shown in Table 2, we found that using {3, 6}-dimensional latent space works well in practice across all embodiments. However, we observed that a 2-dimensional latent space does not perform well for complex agents such as Ant or Humanoid. We hypothesize that, since the PSD objective does not explicitly constrain the circles for each $L$ to lie on the same plane, having additional degrees of freedom allows different circles to occupy different planes. This, in turn, helps stabilize the embedding during training. Conversely, when the latent space is restricted to only 2 dimensions (i.e., a plane), this flexibility is lost, which leads to instability.
>
> **A6-2.** Thank you for your suggestion to add a note that the 2D visualizations are the result of PCA projections! We will include this in the final version. If you have any additional ideas to further improve the reader’s understanding, please feel free to share them, and we will be happy to include them.
>
>
> **Q7. In pixel-based settings, robot limbs are differently colored. Was this necessary for the method to work? An ablation or explanation would clarify its impact.**
>
> **A7.** For symmetric agents like the Ant, limb coloration is a common practice to disambiguate limbs from pixel observations [1]. However, it was not strictly necessary for our specific Ant task (forward locomotion) or for the asymmetric Cheetah. Therefore, while helpful for feature learning, it is not an essential design choice for our method to work.
>
> [1] Park, Seohong, et al. "Ogbench: Benchmarking offline goal-conditioned rl." arXiv preprint arXiv:2410.20092 (2024).
>
> **Q8. Discussion around where PSD may fail or underperform.**
>
> **A8.** PSD may underperform in settings with large, persistent external disturbances (e.g., constant interference from another agent) that make periodic behavior infeasible. We will add this discussion to our limitations section.
>
>
> **Q9. What aspect of vanilla PSD is supervised? Could METRA-PSD also be enhanced with extrinsic rewards for improved downstream performance?**
>
> **A9.** PSD itself is unsupervised as its objective is task-agnostic. The METRA-PSD setup is described as "fully unsupervised" because it combines two unsupervised methods without external supervision. As you suggest, extrinsic rewards can optionally be added to this framework to bias exploration, but this is a separate design choice.
>
> ---
> Thank you again for your valuable review. We hope our responses have addressed your questions.

---

> > ### Comment · Reviewer_N2n9 · 2025-08-04
> > **Response to Rebuttal**
> >
> > Thank you for the detailed response. Your clarifications helped improve my understanding of the method and parts of the evaluation. The additional experiments with more random seeds and hyperparameter ablations address my initial concerns regarding the experimental evaluation. While further ablations, particularly on the latent space dimensionality and the role of color for pixel-based inputs, could strengthen the paper, I understand that, given the time constraints and the NeurIPS rebuttal policy, these are not necessary for acceptance. I would therefore increase my rating to “4: Borderline accept.”
> >
> > That said, I still have a couple of questions:
> >
> > 1. The quantitative metric based on frequency width/gap between the maximum and minimum frequencies is interesting. However, it seems specifically tailored to your method. Why is this a meaningful metric for evaluating locomotion skills more generally? More intuitive measures, such as the range of locomotion speeds or positional coverage, might offer a broader perspective on the quality and diversity of learned skills, and could also apply to methods that get stuck in stationary "yoga poses".
> > 2. It is interesting that different seeds discover different gaits for the same period. Do you have any insights into what causes this variation? Could this diversity also be achieved within a single run? For example, might “range rollbacks” help by excluding and later revisiting previously discovered periods?
> >
> > Feel free to expand on your “detailed quantitative results for a wider range of $L$ values”.

---

> > > ### Author Response · Authors · 2025-08-05
> > > **Thank you for your response**
> > >
> > > Dear Reviewer N2n9,
> > >
> > > We are truly grateful that you have raised your score to 4 (Borderline accept), and we deeply appreciate your thoughtful engagement with our work.
> > >
> > > We are happy to address your follow-up questions:
> > >
> > > ---
> > >
> > > **Q1. On the use of frequency width as a quantitative metric.**
> > >
> > > **A1.** Thank you for raising this question.
> > > While general-purpose metrics such as speed or positional coverage are valuable for evaluating overall locomotion, we would like to emphasize that they do not directly capture the primary focus of our work—**the diversity of periodicity**.
> > > For example, policies can exhibit the same average speed while having different periods.
> > > We therefore chose frequency width as it is a specialized metric, intentionally designed to directly and quantitatively evaluate this core contribution of our work.
> > >
> > > **Q2. Question on the cause of gait variation across seeds and the possibility of discovering multiple distinct gaits within a single run.**
> > >
> > > **A2-1.** Thank you for your question. A periodic motion can be generally described by the form $y = a \cdot \sin(\omega t) + b$. Our objective is primarily focused on the **frequency** $\omega$, which corresponds to the period. This leaves other characteristics, such as the **amplitude** $a$ and **bias** $b$, with **more degrees of freedom**. We hypothesize that the variation between seeds arises since the optimization process finds different solutions for the free parameters.
> > > These distinct solutions result in qualitatively different gaits that all satisfy the same periodicity.
> > >
> > > **A2-2.** That’s a great idea. We agree that revisiting previously encountered periods (e.g., via range rollbacks) could implicitly benefit from the gaits already learned by the current policy, potentially enabling the discovery of alternative gaits.
> > >
> > > In addition, we believe that encoding the representation onto a more general geometry, such as the surface of a sphere (instead of a circle), is a promising future direction, as it may allow for representing multiple distinct gaits even within a single period.
> > >
> > > **Q3. Detailed quantitative results for a wider range of $L$ values.**
> > >
> > > **A3.** As mentioned in our earlier response (**A2-3**), we have added quantitative results for a wider range of $L$ values. Following your valuable suggestion, we will include these results in the final version of the paper.
> > >
> > > - Ant
> > >
> > >
> > >     | Relative error [%] | **L = 10** | **L = 20** | **L = 30** |
> > >     | --- | --- | --- | --- |
> > >     | 1-step distance | 0.11 ± 0.02 | 0.09 ± 0.05 | 0.34 ± 0.16 |
> > >     | L-step distance | 0.09 ± 0.02 | 0.08 ± 0.04 | 0.19 ± 0.08 |
> > > - HalfCheetah
> > >
> > >
> > >     | Relative error [%] | **L = 5** | **L = 10** | **L = 15** |
> > >     | --- | --- | --- | --- |
> > >     | 1-step distance | 1.01 ± 0.11 | 0.93 ± 0.28 | 1.11 ± 0.58 |
> > >     | L-step distance | 0.89 ± 0.19 | 0.64 ± 0.24 | 2.18 ± 0.68 |
> > > - Humanoid
> > >
> > >
> > >     | Relative error [%] | **L = 10** | **L = 20** | **L = 30** |
> > >     | --- | --- | --- | --- |
> > >     | 1-step distance | 1.08 ± 0.16 | 0.81 ± 0.21 | 0.51 ± 0.13 |
> > >     | L-step distance | 0.46 ± 0.09 | 0.55 ± 0.18 | 0.39 ± 0.05 |
> > > - Hopper
> > >
> > >
> > >     | Relative error [%] | **L = 10** | **L = 20** | **L = 30** |
> > >     | --- | --- | --- | --- |
> > >     | 1-step distance | 1.21 ± 0.19 | 0.87 ± 0.23 | 0.93 ± 0.12 |
> > >     | L-step distance | 1.52 ± 0.57 | 0.91 ± 0.60 | 0.43 ± 0.13 |
> > > - Walker2D
> > >
> > >
> > >     | Relative error [%] | **L = 10** | **L = 20** | **L = 30** |
> > >     | --- | --- | --- | --- |
> > >     | 1-step distance | 0.88 ± 0.18 | 0.21 ± 0.10 | 0.25 ± 0.14 |
> > >     | L-step distance | 0.68 ± 0.10 | 0.76 ± 0.27 | 0.21 ± 0.08 |
> > >
> > > ---
> > >
> > > We again thank you for replying to our response.
> > >
> > > Your review has helped to strengthen our work, and we truly appreciate your valuable suggestions and all your efforts throughout the review process.

---

### Official Review · Reviewer_DTax · 2025-07-02

**Clarity:** 3
**Significance:** 2
**Originality:** 3
**Rating:** 5
**Confidence:** 3

**Summary:**

The paper presents an unsupervised learning approach for learning locomotion policies that are periodic. This is achieved by constraining the latent space to follow a circular representation, using a set of terms that “encourage” the encoder to converge to such a structure. This way learning can sample from a range of periods that are adaptively bounded as learning progresses. The paper presents examples of learned policies on a range of different embodiments, commonly used for benchmarking, on locomotion tasks with a range of periods, and two downstream tasks. The downstream tasks are evaluated against a set of skill discovery methods.

**Questions:**

- Does the encoder see all the states, including the base position? Would it choose to ignore certain potentially non-periodic dimensions (such as the XY position) in favor of ones like the joint angles? Can it learn to produce periodic motions along those other dimensions too, for example, by moving in circles of various radius in XY space?

- Visualizations on website could be more informative - maybe slow them down, show different colors for skills (like in Figure 8.).

- What is the state space coverage without adding METRA? Does the algorithm on its own encourage state-covering trajectories? In Appendix B.2 it says that it uses an extrinsic velocity reward - is that necessary for learning useful behaviors?

- A comparison of the state space coverage between PSD and the baselines would be good. From Figure 8 it seems that the XY coverage of PSD+METRA is worse than only METRA (based on their own results) - is this the case?

- Similarly, a more in-depth analysis of the trade-offs of PSD+METRA would strengthen the claim that they can easily be combined. Is the loss worse for both methods when combining them?

**Ethical Concerns:**

["NO or VERY MINOR ethics concerns only"]

**Final Justification:**

Solid contribution.

**Limitations:**

Yes

**Paper Formatting Concerns:**

No concerns

**Quality:**

3

**Strengths And Weaknesses:**

Strengths:
-To the best of my knowledge no other work has addressed unsupervised learning of periodic skills so this is a clear novelty and contribution for the work.
-The paper is well written and the approach is clearly presented.
-The approach is evaluated against other current skill discovery approaches and is shown to empirically work better on the benchmarks presented.
-The experiments show that the method does learn periodic skills across a good range and can be combined with extrinsic rewards or other algorithms like METRA, which is another advantage.

Weaknesses:
- Visualizations on website could be more informative - maybe slow them down, show different colors for skills (like in Figure 8.). Examples videos of downstream task behaviors missing.

---

> ### Author Rebuttal · Authors · 2025-07-31
>
> Dear Reviewer DTax,
>
> Thank you for your valuable and insightful comments. We have carefully considered your feedback and prepared our response below:
>
> ---
>
> **Q1. Visualizations on website could be more informative.**
>
> **A1.** Thank you for your thoughtful suggestions! We will incorporate your feedback into the final version of our project page and paper. If you have any additional ideas to further improve the reader’s understanding, please feel free to share them, and we will be happy to include them.
>
> **Q2. Questions on the state space of the encoder.**
>
> **A2-1.** Thank you for your insightful question. The encoder's state representation does not include the global XY base position and consists only of proprioceptive states like joint angles, velocities, and heading angles, as described in Appendix B. This corresponds to the default observation space in standard Gym environments and is a common practice in prior work [1, 2, 3] on robot learning for generating diverse behaviors.
>
> **A2-2.** Furthermore, from the perspective of our downstream task setup, including the global XY position would be particularly problematic. The exploration reward we use encourages consistent forward movement, which makes the XY position an inherently non-periodic states. This would directly conflict with the PSD objective, which requires the latent representation to be identical after $2L$ steps, and would therefore hinder the learning process.
>
> **A2-3.** We verified this with an experiment on the Ant environment. When we removed the exploratory reward, provided the **full state (including the XY coordinates)**, and trained the agent using **only the PSD objective**, we observed that the agent vigorously wiggled in place, making body movements in sync with the period. This behavior is the result of the optimization satisfying our objective, as it makes **both the joint angles and the global XY coordinates periodic.** We will include this result in the final version of our project page.
>
> Therefore, PSD can successfully discover periodic skills even with the full state, which demonstrates the strength of the PSD objective.
>
> [1] Peng, Xue Bin, et al. "Amp: Adversarial motion priors for stylized physics-based character control." *ACM Transactions on Graphics (ToG)* 40.4 (2021): 1-20.
>
> [2] Peng, Xue Bin, et al. "Ase: Large-scale reusable adversarial skill embeddings for physically simulated characters." *ACM Transactions On Graphics (ToG)* 41.4 (2022): 1-17.
>
> [3] Li, Chenhao, et al. "FLD: Fourier Latent Dynamics for Structured Motion Representation and Learning." *The Twelfth International Conference on Learning Representations*.
>
> **Q3. Questions on the state coverage of PSD.**
>
> **A3-1.** We appreciate your constructive comments. We would like to clarify that PSD is not designed to encourage spatially far-reaching, state-covering trajectories. Instead, its primary goal is to achieve **high coverage in the frequency domain** by autonomously discovering a diverse repertoire of skills with different periods, as demonstrated in Figure 3.
>
> **A3-2.** As mentioned in our response to Q2, an external reward is not necessary for learning periodic behaviors. Our method focuses on learning the **style** of motion (*how to behave*). The optional task reward simply provides a clear objective (*what to do*), which allows the various learned periodic styles to be demonstrated and evaluated more effectively.
>
> **Q4. From Figure 8 it seems that the XY coverage of PSD+METRA is worse than only METRA (based on their own results) - is this the case?**
>
> **A4.** Yes, your observation is correct. To quantify this, we measured the state space coverage by discretizing the XY plane (or X-axis for Walker2D) into a 1x1 grid and counting the number of unique cells visited over 32 random skill rollouts, averaged over 5 seeds.
>
> The results confirm that the XY coverage of METRA+PSD is lower than that of METRA alone:
>
> | **State coverage** | **Ant** | **Walker2D** |
> | --- | --- | --- |
> | **METRA only** | 722 ± 32 | 20.3 ± 0.5 |
> | **METRA + PSD** | 337 ± 6 | 19.9 ± 2.1 |
>
> The reason for this trade-off is detailed in our response to your next question (A5).
>
> **Q5.** Similarly, a more in-depth analysis of the trade-offs of PSD+METRA would strengthen the claim that they can easily be combined. Is the loss worse for both methods when combining them?
>
> **A5.** Thank you for your question. Conceptually, METRA and PSD align well, as both methods address temporal aspects of skills: METRA modulates the temporal **direction** ($z$), while PSD modulates the temporal **length** ($L$).
> However, as you suggest, combining them involves a trade-off. While our mutual conditioning approach ($\phi_L(s, z), \phi_m(s, L)$) solves the primary issue of skill collapse, it introduces a secondary trade-off related to representational capacity.
>
> By conditioning each encoder on an additional skill variable, we increase the complexity of its input space. This can slightly diminish the individual performance of each component compared to when it is trained in isolation. Therefore, while the combined policy $\pi(a | s, z, L)$ is significantly more versatile—**offering independent control over both skill direction and period**—this increased modality comes at the cost of a slight reduction in the peak performance of the individual METRA and PSD objectives compared to their standalone counterparts.
>
> ---
>
> Thank you again for your valuable and insightful review.
>
> Also, please let us know if our responses have addressed your questions. If anything requires further clarification, please do not hesitate to let us know.

---

> ### Comment · Area_Chair_BwZi · 2025-08-05
> **Please respond to the authors' response**
>
> Dear reviewer DTax,
>
> Thanks for your reviewing efforts so far. Please respond to the authors' response.
>
> Thanks, Your AC

---

### Official Review · Reviewer_B3mW · 2025-07-03

**Clarity:** 3
**Significance:** 2
**Originality:** 3
**Rating:** 5
**Confidence:** 3

**Summary:**

In this paper, the authors propose Periodic Skill Discovery (PSD), a novel framework for unsupervised skill discovery in reinforcement learning that explicitly targets the learning of periodic behaviors. PSD maps states into a circular latent space, where movement along the circle corresponds to temporal progression, allowing the agent to represent and generate skills with controllable periodicity. The method involves jointly training an encoder that enforces geometric constraints in the latent space and a policy that maximizes a single-step intrinsic reward based on deviation from ideal circular transitions. The experiments evaluate the performance of the proposed method in various locomotion tasks, present ablation studies, and performance comparison with relevant existing techniques.

**Questions:**

1. How many distinct skills does PSD learn in practice? Is the number fixed, or is it implicitly determined by the range of the period variable? How does this number affect the downstream policy’s ability to select useful behaviors?
2. How were the trajectories presented in Figure 5 selected?
3. For the results shown in Table 1, were all methods trained with the same number of environment steps for both skill discovery and downstream high-level policy learning? How were hyperparameters selected across methods? Also, how were the skill sets chosen for the downstream tasks—did the high-level policy have access to the full set of discovered skills, or only a subset?
4. The authors mention that an exploration reward was added to all methods for fair comparison. However, could this uniform addition unintentionally bias the results? Specifically, for methods already incentivizing exploration, could the added reward lead to excessive exploration or destabilize skill quality?

**Ethical Concerns:**

["NO or VERY MINOR ethics concerns only"]

**Final Justification:**

The authors clarified most of my concerns and, provided that some of that discussion makes it into the final version of the paper (if accepted), I believe it is a good contribution to the field.

**Limitations:**

Yes.

**Paper Formatting Concerns:**

No major formatting issues.

**Quality:**

3

**Strengths And Weaknesses:**

The paper introduces a compelling, and novel, idea of capturing periodic behaviors in unsupervised reinforcement learning.

The methodology is clearly presented, with helpful visualizations and geometric intuition that make the latent structure easy to understand.


The authors demonstrate the effectiveness of PSD across multiple MuJoCo tasks and provide comparisons to several baselines, both in terms of skill diversity and downstream task performance.

A particularly interesting contribution is the adaptive sampling of the period variable, which allows the agent to dynamically expand the range of temporal patterns it can express. This aspect has strong practical potential for multi-timescale behaviors. However, the paper does not include ablation studies to isolate its impact. It is unclear how much performance or skill diversity would degrade without this component.

While PSD is well-suited to locomotion tasks with cyclic structure, it remains unclear how well the method would generalize to settings with non-uniform or compositional sequences (e.g., behaviors like “walk → open door → pick up object” in a loop, or partially periodic domains with interruptions).

While Table 1 shows promising improvements in downstream task performance using PSD-learned skills, the standard deviations are large, and results are averaged over only five seeds. Given the inherent variability of RL training, stronger statistical support (e.g., more seeds or hypothesis testing) would strengthen the claims.

---

> ### Author Rebuttal · Authors · 2025-07-27
>
> Dear Reviewer B3mW,
>
> We sincerely appreciate your constructive and insightful comments, which have been invaluable in helping us further strengthen our work.
>
> We prepared our response below:
>
> ---
>
> **Q1. Lack of ablation studies on the use of the adaptive sampling method to isolate its impact.**
>
> **A1-1.** We appreciate your comment. The adaptive sampling method is crucial for autonomously discovering diverse periods without supervision or prior knowledge. It operates by proposing progressively more challenging periods based on the agent's performance (average cumulative sum of $r_{psd}$). Without this method, the agent fails to learn a wide range of periods, even if those wide range of periods are sampled from the beginning of training. This is analogous to curriculum learning, where an agent masters a difficult task by tackling increasingly challenging goals, rather than being presented with the most difficult ones from the beginning.
>
> **A1-2.** To isolate its effect, we conducted an experiment in which a wide range of periods was sampled from the start, but **without the adaptive sampling** method. After training, we rolled out policies conditioned on each period $L$ and compared their sum of $r_{psd}$ with those from our full method using adaptive sampling.
>
> - Ant
>
>
>     |  | **L = 5** | **L = 16** | **L = 28** | **L = 40** |
>     | --- | --- | --- | --- | --- |
>     | **w/ adaptive** | **198.1 ± 1.2** | **188.1 ± 2.1** | 184.7 ± 2.9 | 179.5 ± 4.5 |
>     | **w/o adaptive** | 152.4 ± 3.7 | 187.8 ± 4.2 | **185.5 ± 2.2** | **183.2 ± 1.9** |
>
> - Halfcheetah
>
>
>     |  | **L = 5** | **L = 8** | **L = 11** | **L = 15** |
>     | --- | --- | --- | --- | --- |
>     | **w/ adaptive** | **187.8 ± 2.5** | **179.6 ± 3.4** | **177.3 ± 6.2** | **175.1 ± 2.2** |
>     | **w/o adaptive** | 185.0 ± 2.7 | 177.2 ± 3.3 | 175.9 ± 2.1 | 167.6 ± 6.3 |
>
> - Humanoid
>
>
>     |  | **L = 10** | **L = 26** | **L = 43** | **L = 60** |
>     | --- | --- | --- | --- | --- |
>     | **w/ adaptive** | **373.0 ± 8.4** | **351.9 ± 6.7** | **376.6 ± 9.5** | **385.2 ± 5.9** |
>     | **w/o adaptive** | 367.1 ± 2.5 | 115.5 ± 6.1 | 91.2 ± 14.7 | 100.9 ± 6.9 |
>
> - Hopper
>
>
>     |  | **L = 5** | **L = 20** | **L = 35** | **L = 50** |
>     | --- | --- | --- | --- | --- |
>     | **w/ adaptive** | **351.4 ± 4.9** | 360.1 ± 11.1 | **379.3 ± 9.6** | **353.3 ± 9.1** |
>     | **w/o adaptive** | 309.8 ± 25.6 | **367.1 ± 8.4** | 372.2 ± 2.9 | 337.6 ± 6.7 |
>
> - Walker2D
>
>
>     |  | **L = 5** | **L = 23** | **L = 41** | **L = 60** |
>     | --- | --- | --- | --- | --- |
>     | **w/ adaptive** | **379.6 ± 6.4** | 362.1 ± 12.2 | **366.6 ± 12.1** | **359.5 ± 10.4** |
>     | **w/o adaptive** | 342.7 ± 4.8 | **371.0 ± 4.6** | 345.4 ± 4.7 | 301.8 ± 8.2 |
>
> Based on the results, our adaptive sampling method is crucial for learning a wide range of high-quality skills, as it consistently outperforms the non-adaptive baseline. This is particularly evident at the extremes of the period range, which highlights its effectiveness in discovering a truly diverse skill repertoire.
>
> **Q2. Generalization of PSD to non-uniform or compositional sequences.**
>
> **A2.** Thank you for this insightful question. While the primary focus of this work, similar to other methods in unsupervised skill discovery, is on learning a rich repertoire of low-level skills, we believe that combining PSD with a hierarchical RL (HRL) framework could enable more complex, compositional tasks.
>
> For example, consider a set of low-level policies $\pi_{low}(a|s,z)$, where $z$ represents a low-level skill such as ‘pick’, ‘move’, or ‘place’ and is capable of both periodic and non-periodic behaviors. Based on this, we could then train a high-level policy $\pi_{high}(z|s,L)$ with the PSD objective to encourage it to select skills $z$ as actions for creating complex, looped behaviors. This could be an interesting future direction.
>
> **Q3. Need for stronger statistical support for downstream task performance results.**
>
> **A3.** Thank you for your helpful suggestions. Following your valuable comment, we expanded our experiments from **5 to 10 seeds** to provide stronger statistical support.
>
> | **Downstream task** | **DIAYN** | **DADS** | **CSD** | **METRA** | **PSD (Ours)** |
> | --- | --- | --- | --- | --- | --- |
> | **HalfCheetah-hurdle** | 0.6 ± 0.5 | 0.9 ± 0.3 | 0.8 ± 0.6 | 1.9 ± 0.8 | **3.8 ± 2.0** |
> | **Walker2D-hurdle** | 2.6 ± 0.5 | 1.9 ± 0.3 | 4.1 ± 1.3 | 3.1 ± 0.5 | **5.4 ± 1.4** |
> | **HalfCheetah-friction** | 13.2 ± 3.4 | 12.4 ± 2.9 | 12.5 ± 3.8 | 30.1 ± 13.1 | **43.4 ± 19.1** |
> | **Walker2D-friction** | 4.6 ± 1.2 | 1.6 ± 0.1 | 5.3 ± 0.3 | 5.2 ± 1.6 | **8.7 ± 1.7** |
>
> For the hurdle and friction tasks, we observed that PSD outperforms the baselines across all embodiments. We hypothesize that the (relatively) high standard deviation of PSD is attributable to the fact that it leverages a more diverse range of periodic behaviors as low-level skills compared to other baselines.
>
> **Q4. Number of distinct skills learned by PSD and its effect on downstream task performance.**
>
> **A4-1.** Thank you for the question. During the training of PSD, we uniformly sample 4 values of $L$ from the range $[L_{min}, L_{max}]$, including both bounds. We found this strategy sufficient, as our model can also achieve the desired periods for interpolated $L$ values after convergence. For more details, please refer to Appendices B.2 and C.2.
>
> **A4-2.** We appreciate your comment. Based on your thoughtful question, we conducted experiments on all downstream tasks, comparing PSD trained with 4 discrete skills and 7 discrete skills within the same bound $[L_{min}, L_{max}]$.
>
> | **Downstream task** | **PSD (w/ 4 skill)** | **PSD (w/ 7 skill)** |
> | --- | --- | --- |
> | **HalfCheetah-hurdle** | 3.8 ± 2.0 | **4.4 ± 1.9** |
> | **Walker2D-hurdle** | **5.4 ± 1.4** | 4.1 ± 1.2 |
> | **HalfCheetah-friction** | **43.4 ± 19.1** | 43.2 ± 13.6 |
> | **Walker2D-friction** | **8.7 ± 1.7** | 8.6 ± 0.6 |
>
> As the results show, increasing the number of discrete skills sampled during training from 4 to 7 does not lead to a significant improvement in downstream task performance. Across all tasks, the performance of both settings is comparable within their standard deviations.
>
> **Q5. How were the trajectories presented in Figure 5 selected?**
>
> **A5.** If your question is about which state components were used to construct the trajectories, we selected the velocities and positions of joints of Ant and Walker2D that best highlight the periodicity. For clear visualization, we normalized these values to lie between -1 and 1. As described in the caption, we also switched the target period $L$ at fixed time intervals within a single episode.
>
> **Q6. Questions on training setup and skill selection for downstream tasks.**
>
> **A6-1.** We trained the low-level skill policy for all methods with the same number of environment steps for each embodiment: 40M for Ant and HalfCheetah, and 80M for Humanoid, Hopper, and Walker2D. We also closely followed the hyperparameters reported in their original papers.
>
> For training the high-level policy for the downstream tasks, we ran 100k episodes across all experiments.
>
> **A6-2.** For all baseline methods, we designed the high-level policy to have access to the full set of discovered skills, matching the skill distribution $p(z)$ used during low-level policy training. For PSD, we uniformly sampled fixed number of $L$ values for $\pi(a|s,L)$ from the range $[L_{min}, L_{max}]$.
>
> **Q7. Questions on the exploration reward.**
>
> **A7.** Thank you for raising this point. Since PSD is designed to learn *how* to behave within a task rather than directly address *what* to do, the specific task setting needs to be predefined when evaluating its performance. For this reason, we added a exploration reward to all methods for fair comparison. Importantly, this exploration reward did not negatively affect the quality of the skills required for downstream tasks. This type of setup where an auxiliary unsupervised reward is combined with the main learning objective is also widely used in prior work on intrinsic motivation [1, 2, 3].
>
> [1] Lee, Youngwoon, et al. "Learning to coordinate manipulation skills via skill behavior diversification." *International conference on learning representations*. 2019.
>
> [2] Li, Jiahui, et al. "Two Heads are Better Than One: A Simple Exploration Framework for Efficient Multi-Agent Reinforcement Learning." *NeurIPS*. 2023.
>
> [3] Bae, Junik, et al. "TLDR: Unsupervised Goal-Conditioned RL via Temporal Distance-Aware Representations." *Conference on Robot Learning*. PMLR, 2025.
>
> ---
>
> Thank you again for your valuable and insightful review. We hope our responses have sufficiently addressed your questions, and we would be happy to provide any further clarification if needed.

---

> > ### Comment · Reviewer_B3mW · 2025-08-05
> >
> > I would like to thank the authors for their thorough reply to my comments.
> >
> > Most of my concerns/questions were clarified, so I will increase my score.
> >
> > I appreciate the number of additional experiments; it does improve the overall quality of the results. I would just like to note that most "shaded regions" are still overlapping, and while 10 seeds is definitely better, I would urge the authors to be careful about their claims based on these results. It indicates that the proposed method performs better, but the statistical significance to claim that, e.g., "the adaptive sampling method is crucial for learning", is not quite there.
> >
> > Once again, thank you for all your work with this rebuttal.

---

> > > ### Author Response · Authors · 2025-08-06
> > > **Thank you for your response**
> > >
> > > Dear Reviewer B3mW,
> > >
> > > Thank you for increasing your score. We're glad our rebuttal addressed your main points. We also appreciate your feedback on statistical significance and will carefully consider this point for the final version.

---

> ### Comment · Area_Chair_BwZi · 2025-08-05
> **Please respond to the authors' response**
>
> Dear reviewer B3mW,
>
> Thanks for your reviewing efforts so far. Please respond to the authors' response.
>
> Thanks, Your AC

---

### Note · Authors · 2025-08-14

Dear AC and Reviewers,

We sincerely thank you for your valuable time and constructive feedback. We are especially grateful for the productive discussion and deeply appreciate that **most reviewers found our rebuttal convincing and agreed to raise their scores.**

---

We propose **Periodic Skill Discovery (PSD)**, a novel framework that enables the agent to learn a rich set of periodic behaviors across varying timescales.

The key contributions of our work are as follows:

- We propose a novel unsupervised skill discovery objective that leverages a circular latent space to autonomously discover periodic behaviors across multiple timescales.
- We introduce an adaptive sampling method that enables the discovery of a dynamically feasible and maximally diverse set of periods, and by leveraging these learned skills, PSD enables agents to solve complex downstream tasks more effectively.
- By encoding temporal distance rather than relying on specific state representations, PSD can discover various periodic behaviors even in pixel-based environments.
- PSD can be combined with the existing skill discovery method METRA, thereby expanding the range of learnable behaviors.

Reviewers noted that our method is **novel, well-motivated, and clearly presented**. During the rebuttal, we incorporated additional analyses—including ablation studies, quantitative evaluation across multiple seeds, and further downstream task experiments—which further clarified the statistical significance and robustness of our findings.

---

Once again, we thank the AC and Reviewers for their thoughtful evaluation and consideration.

---

### Decision · Program_Chairs · 2025-09-17

**Decision:**

Accept (poster)

**Comment:**

This paper proposes an unsupervised skill discovery approach tailored to environments / robots where periodicity is helpful to solving relevant tasks, e.g. locomotion. The method trains an encoder of the desired periodic variables to encode to a time-profiled circular latent space through a novel skill discovery objective. The paper claims that the discovered skills are predictive over multiple horizons, can be applied to image observations, and can be combined with another skill discovery method.

Reviewers perceived a variety of strengths, including the novelty, good motivation, and clear presentation.

Reviewers perceived a few weaknesses, including limited application to locomotion tasks, and that the method requires prior knowledge of what subset of the observation should be periodic (e.g. a global position XY in the observations was found to hurt performance).

Overall the paper appears to soundly demonstrate its claimed contributions and these claimed contributions are significant. I concur that the main weakness in the method is that it requires prior knowledge of what aspects of the observation should be periodic, a method that addressed this weakness would help enhance the applicability to other domains beyond locomotion.